# ALFA-K: Local adaptive mapping of karyotype fitness landscapes

Richard J. Beck, Tao Li ⓘ & Noemi Andor ⓘ ✉

Despite its critical role in tumor evolution, a detailed quantitative under-standing of the evolutionary dynamics of aneuploidy remains elusive. Here we introduce ALFA-K (Adaptive Local Fitness landscapes for Aneuploid Kar-yotypes), a method that infers chromosome-level karyotype fitness landscapes from longitudinal single-cell data. ALFA-K estimates fitness of thousands of karyotypes closely related to observed populations, enabling robust predic-tion of emergent karyotypes not yet experimentally detected. We validate ALFA-K's performance using synthetic data from an agent-based model and empirical data from in vitro and in vivo passaged cell lines. Analysis of fitted landscapes suggests several key insights: (1) Whole genome doubling facil-itates aneuploidy evolution by narrowing the spectrum of deleterious copy-number changes; (2) Environmental context and cisplatin treatment sig-nificantly modulate the fitness impact of these changes; (3) Fitness effects of copy-number changes depend on parental karyotype; and (4) Chromosome mis-segregation rates strongly influence the predominant karyotypes in evol-ving populations.

Losses and gains of entire chromosomes or large sections thereof, known as aneuploidy, are a defining feature of solid tumors[1,2]. This aberrant state arises from an ongoing dynamic process termed chro-mosomal instability (CIN), which stems from diverse mechanisms including mitotic segregation errors, replication stress, and structural chromosome damage[3]. While CIN can also manifest as structural aberrations, this work focuses primarily on whole-chromosome CIN and the resulting numerical aneuploidy state[3], as these whole-chromosome changes affect more of the cancer genome than any other genetic alteration[4]. Present in an estimated 90% of solid tumors, aneuploidy modifies the copy number of many genes, thereby altering cellular phenotype through correlated changes in RNA expression and protein production[5,6]. Aneuploidy provides a substrate for tumor evolution[1,7,8], often enriching chromosomes with oncogenes while deleting those with tumor suppressor genes[9].

The factors that explain aneuploidy patterns in cancers are not limited to the density of driver or suppressor genes on a particular chromosome. Aneuploidy is usually detrimental to cell fitness[10], in the first instance due to proteins such as P53, which cause apoptosis or cell cycle arrest in response to chromosome missegregations[11]. According

to the gene dosage hypothesis, aneuploidy can also reduce fitness by upsetting the balance of protein levels within cells, leading to negative effects such as impaired formation of stoichiometry-dependent pro-tein complexes, or protein aggregates that overwhelm protein quality-control mechanisms[10,11]. In addition to these intracellular effects, the environmental context plays a role in sculpting karyotype[4], since the specific pressures of an environment will determine whether the fit-ness advantages of a particular copy number alteration (CNA) out-weigh the costs. Evidence for the role of environment in determining karyotype includes the selective advantage of particular karyotypes under stressful conditions in yeast[5] and distinct patterns of aneu-ploidies between cancer types[12–14]. Genomic context also plays a role in sculpting karyotype because a given CNA may only be favorable if other mutations or CNAs are already present within the cell. Support for the role of genomic context comes from findings that CNAs lacking independent prognostic value can predict survival in combination[15], as well as from the reproducible temporal ordering of CNAs in patient-derived xenograft and organoid models[16,17].

Aneuploidy remains difficult to study, for reasons which include the difficulty of experimentally inducing aneuploidy and the difficulty

H. Lee Moffitt Cancer Center and Research Institute, Integrated Mathematical Oncology, Tampa, Florida, USA. ✉e-mail: noemi.andor@moffitt.org

of distinguishing the effects of aneuploidy from those of chromosomal instability, the process which causes aneuploidy[4]. In silico models will be an important tool to further our knowledge of aneuploidy. Gusev and colleagues developed the first model describing whole-chromosome missegregations[18,19]. This model laid the mathematical foundations for describing segregation errors and explained patterns of aneuploidy in experimental data as a consequence of variable chromosome missegregation rate. A limitation of this model was that a fitness landscape defining the effect of aneuploidy on cell fitness was not considered, beyond a constraint that cells losing all copies of any chromosome were not considered viable. This limitation was later addressed by others who assumed that the fitness effect of changing the copy number of a particular chromosome was dependent on the number of oncogenes or tumor suppressor genes expressed on that chromosome[20,21]. In these models, cell fitness could be increased by gaining additional copies of chromosomes with many oncogenes or losing copies of chromosomes with many tumor suppressor genes. These models predicted an optimal missegregation rate (in the sense of minimizing total cell death) that matched experimental observations, and resulted in a near-triploid karyotype that is frequently observed in tumor cells. More recent work explored alternative fitness assumptions, considering not only driver gene density but also stabilizing selection based on overall gene abundance or a hybrid of both approaches, while using simulations to infer CIN rates[22]. All of these previous models of aneuploidy fail to account for how environmental influences and genomic background impact the relationship between karyotype and fitness. This is perhaps unsurprising, since the vast number of possible karyotypes is challenging enough to map even without the additional variability introduced by these contexts. However, the burgeoning quantity of single-cell copy number data[16,23] now permits tracking of subclonal evolution at unprecedented resolution, providing the potential to refine our understanding of the relationships between karyotype and cellular fitness.

Fitness landscapes represent a mapping from genome to cellular fitness. Charting these landscapes for karyotypes is particularly challenging due to the vast number of possible states—over $10^{19}$ karyotypes when considering up to eight copies per chromosome. Here, we introduce ALFA-K, to our knowledge, the first method to directly chart local regions of karyotype fitness landscapes from single-cell copy number data. After validating ALFA-K with synthetic data from an agent-based model of chromosome missegregations, we further confirmed its predictive accuracy using empirical data from P53-deficient cell lines showing extensive subclonal evolution. Our analysis yielded several insights, highlighting roles for whole genome doubling, in vivo selection pressures, cisplatin treatment, and chromosome missegregation rates in shaping the topology of karyotype fitness landscapes and cellular evolutionary trajectories.

## Results

### ALFA-K predicts karyotype evolution of variable speed and complexity

We developed a framework to infer **A**daptive **L**ocal **F**itness landscapes for **A**neuploid **K**arotypes (ALFA-K). The method utilizes longitudinal data from evolving cell populations (Fig. 1a), where single cells are analyzed via sequencing to determine their specific karyotypes. This allows for tracking temporal changes in the frequency of common karyotypes within the population (Fig. 1b). ALFA-K assumes that fitness is intrinsically determined by karyotype, equating to the net growth rate, and that this fitness landscape is static over the observation period (conceptualized in Fig. 1c). Based on this, fitness estimates for common karyotypes are derived directly from their observed frequency changes over time. These initial estimates are then extended to rarer, related karyotypes through Gaussian process regression, thereby constructing an inferred local fitness landscape covering the explored karyotypic space (Fig. 1d). The framework uniquely accounts

for the fact that the fitness consequences of missegregation (MS) can vary depending on the specific karyotype background of the missegregating cell (Supplementary Note 1).

We first validated ALFA-K on synthetic datasets. We set up agent-based model (ABM) simulations evolving on Gaussian random-field fitness landscapes (Fig. 1e), then trained and evaluated ALFA-K performance on simulated data across a period of active population evolution (Fig. 1f). Accuracy increased on smoother terrains and when more longitudinal passages were available, whereas systematic error stemmed mainly from inaccurate fitness estimates for well-sampled ("frequent") karyotypes (Fig. 2). ABM simulations using fitted fitness landscapes as inputs reproduced clonal dynamics (Figs. 1g, 5), supporting subsequent application to longitudinal single-cell experiments.

To evaluate ALFA-K on experimental data, we fitted the model to longitudinal single-cell karyotype profiles from two breast-cancer cell lines and four patient-derived xenograft (PDX) transplant series cultured with or without cisplatin[23] (Fig. 2a, see Supplementary Note 4.1 for details). Because ground-truth fitness landscapes are unknown for experimental data, we evaluated internal consistency using leave-one-out cross-validation (CV) scores, where higher values (max 1.0) indicate better fit. Here, a *trajectory* is the ordered sequence of passages obtained from a single lineage (e.g., $A \rightarrow B \rightarrow C$) and its nested subsequences (e.g., $A \rightarrow B$, $B \rightarrow C$); each trajectory yields one ALFA-K landscape. Across the six independent lineages, this definition produced **275** candidate trajectories. In line with in silico tests, CV scores generally increased with the number of training passages used (Fig. 2b). ALFA-K landscapes obtained on trajectories of size four or more resulted in a median CV score of 0.48 (cluster-bootstrap 95% CI 0.11−0.57). Across all fitted landscapes with a positive CV score, ALFA-K returned fitness estimates for 272,317 unique karyotypes.

We then tested ALFA-K's ability to forecast the karyotype distribution of the subsequent passage. We retained fits with positive cross-validation score and at least one downstream passage, keeping only the highest-scoring fit per final timepoint so that each defined a distinct test instance ($N = 35$ fitted lineages). To evaluate predictive performance, we used the angle metric, which measures the similarity between two evolutionary trajectories (an angle of 0° indicates identical paths). We calculated the angle between the observed evolutionary changes and those predicted by either ALFA-K or experimental sister passages. Both sister-passages (median-of-lineage-medians 79. 0°, cluster bootstrap $p = 0.0076$) and ALFA-K predictions (median-of-lineage-medians 71. 2°, cluster bootstrap $p < 0.0001$) were significantly more aligned than random orientations. Comparisons between unrelated lineages followed the null distribution (median-of-pair-medians 90.4°, parametric double-cluster bootstrap $p = 0.80$), confirming that the model captures lineage-specific evolutionary signal (Fig. 2c). Furthermore, ALFA-K recovered the overall scale of evolutionary change; predicted Wasserstein distances between consecutive passages correlated well with the observed distances (Spearman $\rho = 0.68$, Pearson $r = 0.72$, Fig. 2d).

Next, we compared ALFA-K forecasts to a static "no-evolution" baseline, where the predicted state is simply the last observed state. The fraction of sub-lineages where ALFA-K outperformed this baseline was highest immediately after training and declined over 15−25 days (Fig. 2e). We specifically evaluated performance at the next measured timepoint. We compared how often ALFA-K outperformed the static baseline versus how often experimental sister passages (biological replicates) did so (Fig. 2f, g). Sister-passage predictions outperformed the baseline in 56% of cases, whereas ALFA-K forecasts did so in 36%. However, these success rates are not directly comparable because the test sets differed: ALFA-K was evaluated on sub-lineages with high internal consistency scores, while the sister-passage comparison was restricted to lineages where experimental forks were available (Fig. 2h). Overall, the rates at which both methods surpassed the

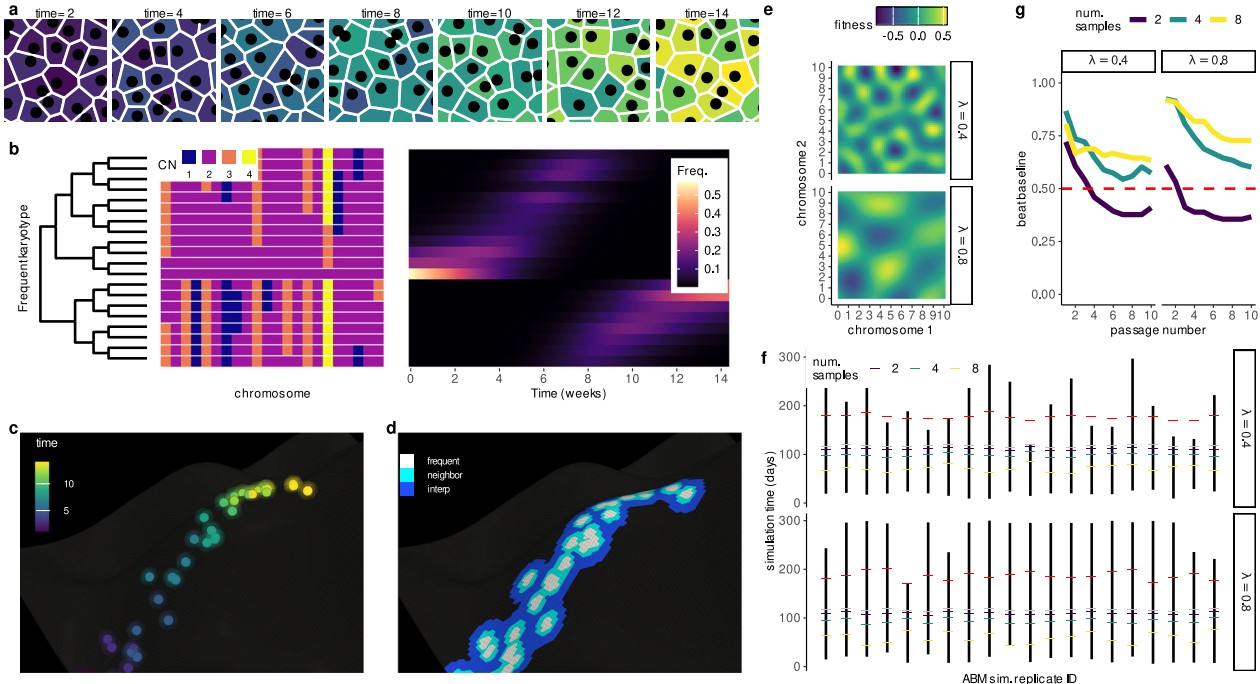

**Fig. 1 | Inferring aneuploid fitness landscapes with ALFA-K. Conceptual overview. a** Schematic representation of an evolving cell population passaged longitudinally over multiple timepoints (panels). Individual cells are colored based on the time point at which their specific karyotype first emerged in the population. **b** Karyotypes (rows) are determined from single-cell sequencing data (Supplementary Note 4.1). The left heatmap displays inferred chromosome copy numbers (fill color) for various detected karyotypes. In the right heatmap, the frequency (fill color) of distinct karyotypes (y-axis) across different timepoints (x-axis) is shown. **c** Conceptual visualization of karyotype evolution on a fitness landscape. Each point represents a unique karyotype, positioned according to a 2D projection of its high-dimensional state (x and y axes), with fitness indicated by height (z-axis). Points are colored by their time of first emergence, corresponding to panel (**a**). **d** Using the same representation as (**c**), but highlighting instead the region where fitness estimates are made by the pipeline. Fill color within the charted region indicates the

stage of the inference process used to estimate fitness for that specific karyotype (e.g., direct frequency-based estimation vs. Gaussian process regression, see Supplementary Note 1). **Validation of forecasting performance. e** Two example Gaussian-random-field (GRF) fitness landscapes illustrate how increasing the wavelength (λ) alters topology. **f** Overview of ABM sampling strategy. Agent-based simulations incorporating MS-driven karyotype changes were run on the GRF landscapes in (**e**); the resulting karyotype counts served to train and validate ALFA-K. Black bars indicate the longest continuous fitness-increasing interval per simulation; colored ticks mark the initial timepoint for each training window; gray ticks indicate the final training timepoint, and red ticks mark the prediction horizon. **g** Fraction of ABM simulations whose forecasts outperform a Euclidean "no-evolution" baseline (see Supplementary Note 3.1), evaluated on unseen training data after excluding landscapes with poor internal consistency. Source data underlying these plots are available in the ALFA-K repository.

baseline were modest, underscoring the difficulty in forecasting subtle clonal shifts over these short intervals.

These results suggest that ALFA-K captures directional and quantitative trends in karyotype evolution, offering predictive power that approaches replicate-passage comparisons, particularly in the near term. However, the modest rates at which forecasts outperformed the static baseline (Fig. 2e–g) highlight the inherent challenge of making accurate, fine-grained predictions of clonal population dynamics over typical experimental timescales.

## ALFA-K predicts the emergence of novel karyotypes

Prior work by Salehi et al.[23] demonstrated that the dynamics of karyotype-defined subpopulations can be extrapolated over time, but their method was limited to karyotypes already observed. In contrast, we asked whether ALFA-K could forecast the emergence of previously unobserved karyotypes–i.e., genotypes that had not yet appeared but could arise based on their fitness and location in the karyotype space.

To evaluate the predictive performance of ALFA-K for previously unobserved karyotypes, we focused on the same subset of 35 fitted karyotype landscapes used in earlier predictive analyses (Fig. 2c–f), selected for their availability of future time points that allow assessment of novel karyotype emergence. Clonal interference frequently arises in these landscapes, where multiple fit karyotypes co-exist and compete for dominance over time. As shown in Fig. 3a, these dynamics can obscure which karyotypes will ultimately expand, especially when

several are predicted to have similarly high fitness. Capturing the effects of such competition is therefore central to ALFA-K's ability to correctly forecast emergent karyotypes. The selected lineages were distributed across five cell lines as follows: p53 k.o – 7; SA1035 – 3; SA532 – 6; SA535 – 6; SA609 – 13.

Let Θ denote the set of karyotypes with fitness estimates from ALFA-K, ζ the subset of Θ observed in a given longitudinal sample. Then we would like to predict Ψ (the subset of Θ that will be present in a future sample) (Fig. 3a, b). In particular, we wish to predict which new karyotypes will emerge in the next sample, $'ζ ∩ Ψ$ (Fig. 3c). The probability of any novel karyotype actually emerging presumably depends both on its fitness and its number of neighbors in the preceding generation. Therefore, for each member of $'ζ$, we computed the fraction of karyotypes in ζ that were between 1–5 missegregations distant (Fig. 3d). These were used as variables which, together with the fitness estimate ($f$), were used to predict whether the karyotype would emerge. For prediction, we used binomial logistic regression, then assessed whether each predictor variable was significantly correlated with the response variable. As expected, the fraction of ζ that were distance-1 neighbors ($d_1$) was a very significant predictor of novel karyotype emergence (Fig. 3e, f). The fitness estimates from ALFA-K also contributed significantly to the prediction, being significant in 15/45 tests (Fig. 3e) and the most significant predictor in 7/45 tests (Fig. 3f). These results indicate that ALFA-K fitness estimates can help predict the emergence of new karyotypes.

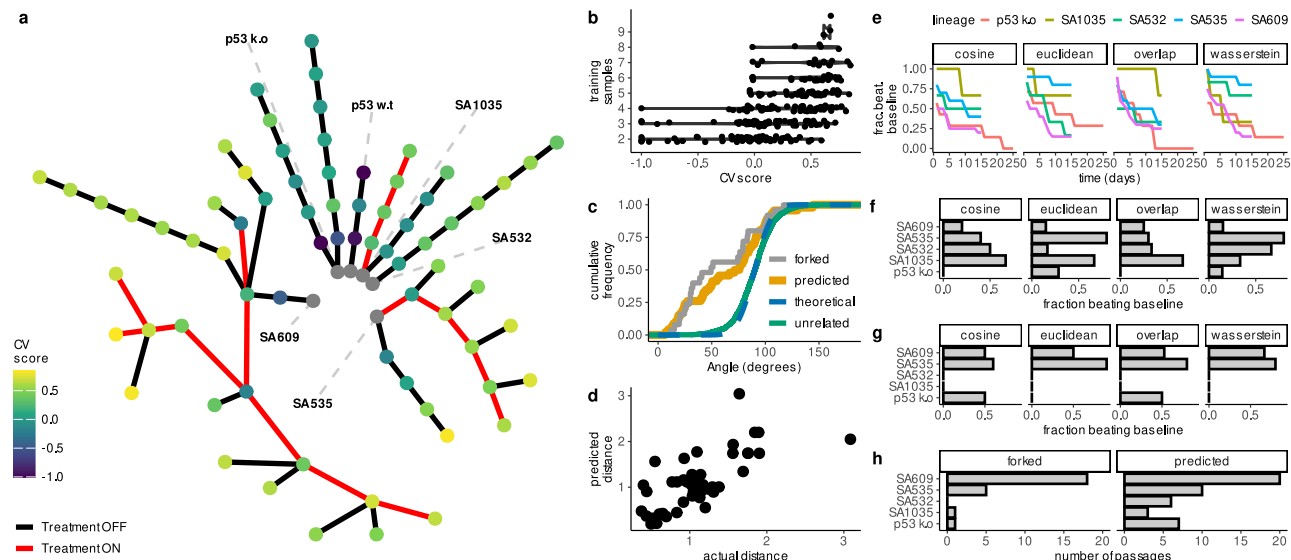

**Fig. 2 | Validation of ALFA-K on longitudinal single-cell data. a** Lineage graph showing experimental passages (nodes) colored by ALFA-K CV score, used here as a proxy for model fit in the absence of ground truth fitness. Red edges indicate intervals of cisplatin treatment. To select inputs for forecast validation (**c–g**), we defined sublineages as any passage plus a preceding history; for each potential endpoint, the history length and frequent clone threshold maximizing the CV score were chosen. Sublineages with CV score ≤ 0 were excluded, yielding 35 sublineages for analysis. **b** CV score generally improves with the number of passages used for training. **c** Cumulative frequency distribution functions (CDFs) of angle metrics comparing forecasts to ground truth. ALFA-K forecasts (predicted) show similar directional accuracy to comparisons between sister passages (forked). Both are significantly more aligned with the true direction than the random-orientation null (theoretical), which is matched by comparisons between unrelated lineages. **d** Predicted vs. observed Wasserstein distances between consecutive passages. **e** Fraction of the 35 selected sub-lineages where ALFA-K forecasts outperform a static "no-evolution" baseline, plotted against time since training for all metrics. **f** Fraction of ALFA-K forecasts outperforming the static baseline at the specific forecast horizon corresponding to the next measured passage. **g** Fraction of sister passages which "beat baseline", evaluated on available forked passages. **h** Number of passages available for metric evaluation corresponding to (**f–g**). Source data underlying these plots are available in the ALFA-K repository.

## Karyotypic background determines fitness effects of CNAs

We set out to quantify how experimental context (in vitro vs. PDX) and cisplatin exposure reshape the fitness landscapes inferred with ALFA-K. To ensure statistical independence, we implemented a bootstrap procedure, repeatedly sampling sets of evolutionary trajectories with no shared passages (Fig. 4a). For each of 200 bootstrap replicates, we re-estimated all model coefficients and deemed an effect significant ($p < 0.05$) when its 95% bootstrap confidence interval did not include zero. For each frequent karyotype in every fitted landscape, we calculated the 44 one-MS-step fitness effects ($\Delta f$) (Fig. 4b). Visualizing the raw $\Delta f$ distributions for a typical bootstrap replicate revealed only minor differences in the median but substantial broadening of the distribution tails (Fig. 4c). To quantify the visual pattern, we used two GLMMs—one Gaussian on the log-variance of each karyotype's $\Delta f$ profile, the other log-Gamma on the absolute $\Delta f$ across all karyotypes and landscapes. These approaches examine complementary aspects of the same underlying landscapes: the log-variance captures how widely fitness effects vary within local karyotypic backgrounds, while the log-absolute $\Delta f$ reflects the typical size of individual mutational steps. The analysis revealed that the coefficients for both PDX context and cisplatin treatment were positive and significant across all models. The 95% confidence intervals, derived from the distribution of estimates across all bootstrap replicates, were entirely above zero, confirming that both conditions significantly increase fitness effect magnitude and variance (Fig. 4d). Together, the elevated variance and larger individual fitness effects suggest that PDX growth and cisplatin exposure increase the availability of both advantageous and deleterious CNA steps. Stronger fitness effects are expected to accelerate clonal expansions and extinctions, leading to greater shifts in population composition between passages. Consistent with this, clone–composition changes between successive passages were more pronounced under these conditions (Fig. 4e): overlap coefficients

decline from in vitro controls to PDX controls and drop further after cisplatin, indicating that compositional shifts are larger when $|\Delta f|$ and variance are high.

To quantify how similarity between $\Delta f$ profiles depends on karyotype distance and experimental context, we fitted a linear mixed model with the Pearson correlation between $\Delta f$ profiles for pairs of karyotypes as the response variable. Relative to a reference of cisplatin-treated pairs with identical karyotype, drawn from different cell lines—whose correlation was indistinguishable from zero (Fig. 4f)—three patterns emerged. First, membership in the same inferred fitness landscape markedly increased correlation. Second, within-landscape correlation decreased sharply with growing karyotype distance, confirming a distance-decay relationship. Third, karyotype pairs that evolved in parallel from the same cell line showed a modest uptick in correlation, but this effect did not reach significance in the bootstrap analysis. No other covariates, including shared treatment background, had a detectable influence. Taken together, these results underscore that physical proximity in karyotype space is an important determinant of CNA-fitness profile similarity.

We next asked how whole-genome doubling (WGD) alters both the pace of aneuploidy accumulation and its predicted fitness consequences in our two p53 k.o. lineages. Modeling the passaging trajectories revealed that WGD$^+$ populations accumulated more chromosome alterations resulting in a significantly higher level of aneuploidy compared to WGD$^-$ populations (Fig. 5a, non-linear saturation model, WGD effect on asymptote Wald test $P < 10^{-6}$). ALFA-K fitness estimates generally increased with the number of chromosome alterations in both WGD$^-$ and WGD$^+$ populations (Fig. 5b). However, the relationship displayed distinct visual patterns: mean WGD$^-$ fitness appeared to rise steeply for the first 2–3 alterations before leveling off, while mean WGD$^+$ fitness continued to increase across the broader range of aneuploidy observed in these subclones.

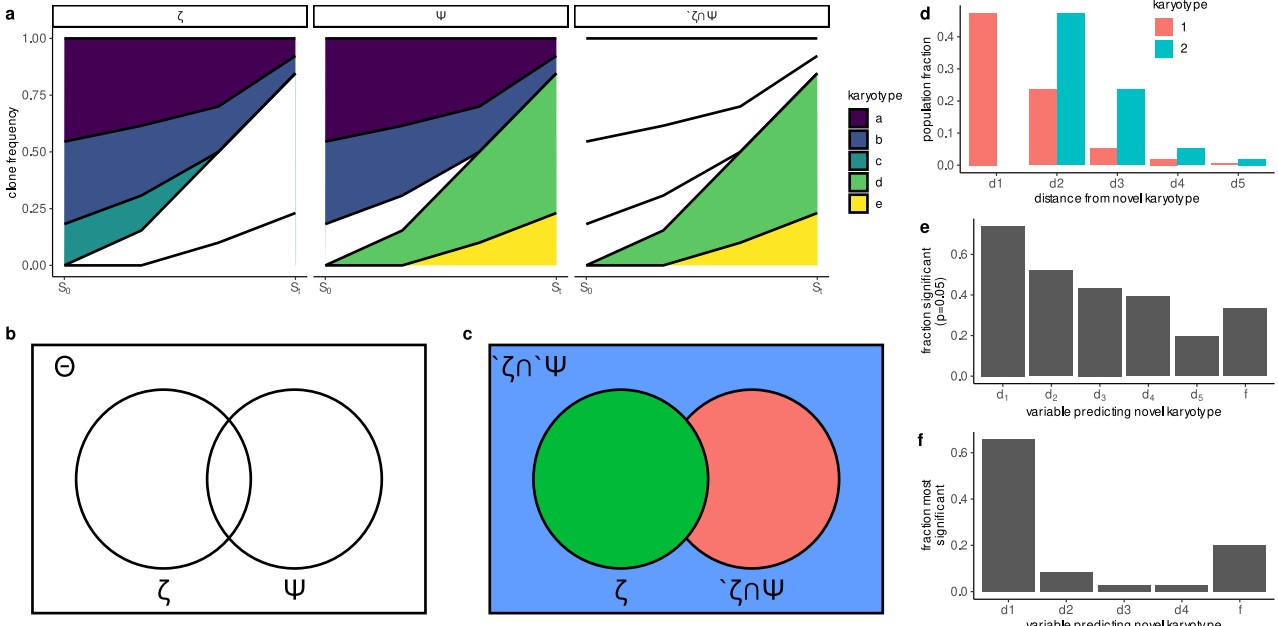

**Fig. 3 | ALFA-K predicts emergence of novel karyotypes. a** Clonal interference—simultaneous expansion of multiple advantageous karyotypes that compete and slow one another's fixation—is illustrated for 5 example karyotypes. Clone group (columns) is categorized as illustrated in (**b**, **c**). $S_0$ and $S_t$ indicate present and future sample timepoints, respectively. **b** Venn diagram representing all karyotypes in the fitted landscape ($\Theta$), the subset observed in the latest sample ($\zeta$), and the subset that will be present in a future sample ($\Psi$). **c** $\Theta$ is separated into 3 disjoint subsets. The aim is to predict $'\zeta \cap \Psi$. **d** Distance feature vector assigned for two example karyotypes: $\mathbf{k}_1, \mathbf{k}_2 \in '\zeta \cap \Psi$. $d_i$ is the fraction of karyotypes in $\zeta$ that are $i$ missegregations away from $\mathbf{k}_1/\mathbf{k}_2$. **e**, **f** Comparing the contribution of $f$ (the ALFA-K estimated karyotype fitness) to that of other variables as predictors of novel karyotypes $\mathbf{k}^* \in '\zeta \cap \Psi$. Fitness estimates used here are computed using data prior to the point of emergence being predicted to avoid look-ahead bias. **e** Fraction of fitted lineages in which each variable was significant ($P < 0.05$). **f** Fraction of fitted lineages in which each variable was most significant. Significance in (**e**, **f**) was determined by the two-sided Wald test P-value of the test statistic for each fitted parameter. Source data underlying these plots are available in the ALFA-K repository.

Because ALFA-K restricts fitness predictions to karyotypes within two mis-segregations of the observed frequent clones ("Methods"), the upper bound on "chromosomes altered" (Fig. 5b) reflects how far each sub-population actually explored karyotype space. Finally, the empirical cumulative distributions of one-MS-step fitness effects (Fig. 5c) are shifted toward less deleterious outcomes in WGD$^+$ subclones (permutation KS test on fit-level averages, empirical $p = 0.03$), suggesting that WGD not only accelerates karyotypic diversification but also reduces the pool of fitness-decreasing aneuploid states.

**Missegregation rate influences karyotype dominance**
In most circumstances, the fittest karyotype is expected to eventually dominate the population. However, we asked whether this outcome could be altered simply by changing the missegregation rate. Cells with similar karyotypes that occupy the same fitness peak can be conceptualized as a "quasispecies"[24], and under the quasispecies framework, sufficiently rapid mutation can cause populations to drift off narrow fitness peaks—a phenomenon known as the error threshold[24]. We therefore considered two ways that elevated missegregation might erode dominance (Fig. 6a): first, quasispecies containing many high-fitness karyotypes incur smaller fitness penalties from continual missegregation events, because the resulting progeny remain relatively fit; second, low-ploidy quasispecies experience fewer errors per division, allowing for greater population stability. If the fittest karyotypes have few fit neighbors or are of high ploidy, they may ultimately be displaced under high missegregation rates.

To identify rate-dependent switches, we built approximate transition matrices over the charted region of each ALFA-K–fitted landscape and determined the steady state distribution of karyotypes across missegregation rates (Supplementary Note 1.6.1). As an initial screen, we searched for landscapes in which the most frequent

karyotype (used as a proxy for its quasispecies) differed across missegregation rates. 28 out of 35 tested sublineages met this criterion (Fig. 6b). Two examples were explored in more depth. For both p53 k.o. A and SA532, we retained the 400 most abundant karyotypes and assigned them group membership: group "x", those with abundance negatively correlated to missegregation rate, and group "y" those with abundance positively correlated to missegregation rate. The aggregate frequency of each group shows a clear critical threshold whereupon group dominance within the population flips (Fig. 6c). Moreover, UMAP projections in karyotype space show clear separation between the groups, supporting their interpretation as quasispecies (Fig. 6d, e). Histograms of fitness (Fig. 6f) reveal that in both examples, the low-rate dominant group contains the fittest karyotypes. For p53 k.o. A, the high-rate dominants have lower ploidy, whereas for SA532 the selected x/y groups have similar ploidy; by contrast the high-rate dominants enjoy a neighborhood size advantage (Fig. 6g) - supporting the assignment of each cell line to the example scenarios (Fig. 6g). These results demonstrate that missegregation rate can interact with both karyotype fitness landscape shape and ploidy to determine clonal dominance in culture.

## Discussion
To our knowledge, ALFA-K is the first attempt to characterize local regions of karyotype fitness landscapes. We and others have previously modeled karyotypic evolution through the process of missegregation, employing various assumptions about the fitness associated with specific karyotypes. These assumptions range from considering all karyotypes equally fit[19], associating fitness with the density of driver or suppressor genes on each chromosome[20,21], to correlating fitness negatively with deviations from a euploid state[25]. These models, however, have typically neglected the context in which

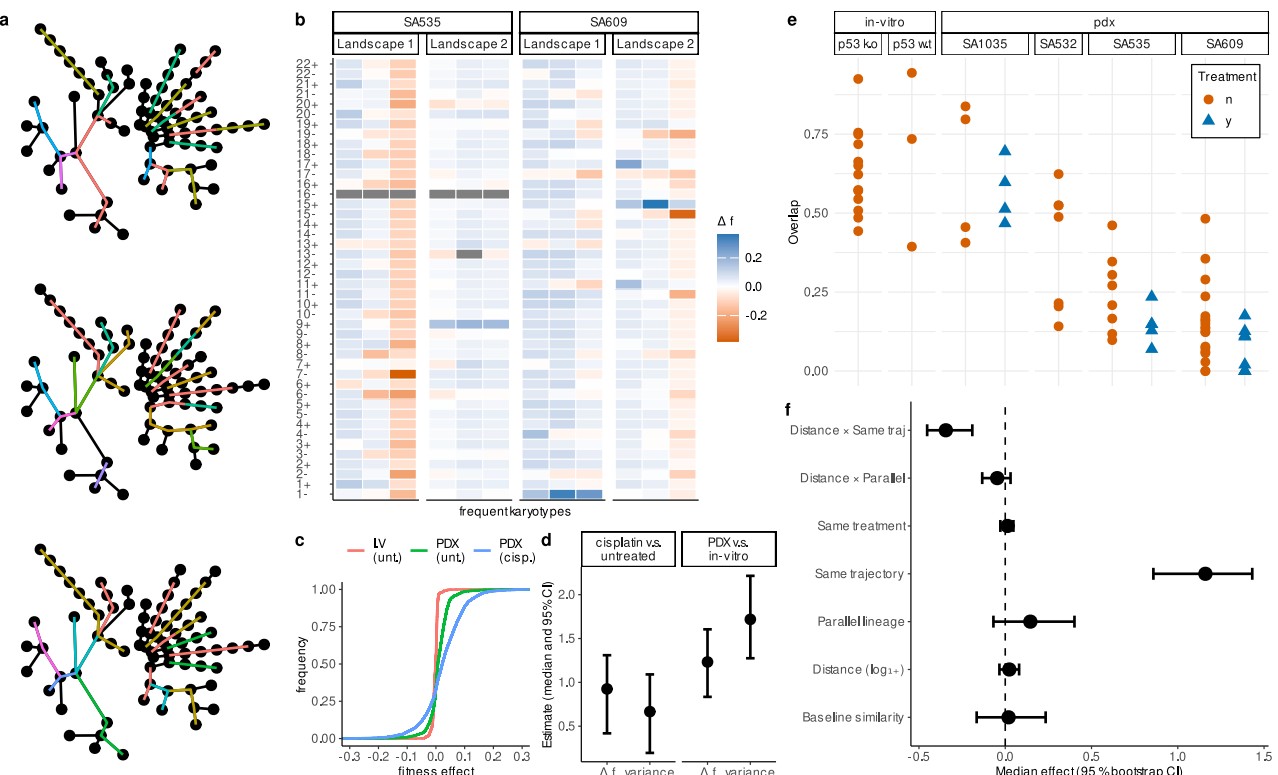

**Fig. 4 | Parent cell karyotype and treatment context shape CNA fitness effects.** **a** Radial dendrograms of all sequenced passages. Colored segments indicate different sets of non-overlapping evolutionary trajectories used within different bootstrap replicates. **b** For each karyotype, the fitness change associated with gaining or losing each chromosome was predicted by ALFA-K. The resulting vectors (Δf profiles) for sample karyotypes are shown in each column. **c** Empirical cumulative-distribution functions of Δf for the three analysed conditions, from a single bootstrap replicate. **d** Estimated effects of experimental context and treatment on the fitness landscape. Points show the median coefficient estimates from 200 bootstrap replicates, with bars indicating the 95% confidence intervals. Effects are shown for both the magnitude of fitness changes (Δf) and their variance. **e** Abundance-overlap coefficients between consecutive passages, plotted by cell line, colored and shaped by treatment. Lower overlap signifies larger compositional changes from one passage to the next. **f** Factors influencing the Pearson correlation of Δf profiles between karyotype pairs. The plot shows the median coefficient estimates (points) and 95% confidence intervals (bars) from a bootstrapped linear mixed model with 200 bootstrap replicates. The "baseline" estimate represents the similarity for karoytypically identical, cisplatin-treated karyotype pairs from different cell lines. The slope term (Distance) quantifies how correlation changes with (log transformed) karyotypic distance, while its interaction coefficients indicate how that distance-decay is modified by pair type or treatment. All remaining categorical coefficients represent additive shifts from the baseline. Confidence intervals that do not intersect zero mark statistically significant effects (*p* = 0.05). Source data underlying these plots are available in the ALFA-K repository.

karyotypic fitness is embedded—factors such as genetic background, tumor microenvironment, and immune interactions profoundly shape the fitness landscape. The vast number of possible karyotypes further complicates the reconstruction of fitness landscapes. Our mathematical model introduces the flexibility needed to begin reconstructing adaptive fitness landscapes, allowing us to extrapolate the fitness of thousands of karyotypes based on the dynamics of just a few subclones.

Whilst the fitness landscapes reconstructed by ALFA-K are not mechanistic, they can be leveraged to provide mechanistic insights into the evolutionary dynamics of the underlying cell populations. It has been observed that cells undergoing WGD exhibit a higher rate of chromosomal alterations than non-WGD cells[26]. The reasons behind this accelerated evolution remain unclear: it could be due to either an increased rate of CNA generation or a heightened tolerance to the deleterious effects of CNAs in WGD cells. We explored this question using an empirical dataset from a cell line that underwent WGD during passaging[23]. Our analysis of the fitness landscape in this line suggests that WGD karyotypes display fewer deleterious CNAs. This supports the notion that WGD may facilitate chromosomal evolution by enhancing cellular robustness to CNAs.

We assessed whether CNA fitness effects generalize across different genomic and experimental contexts by comparing the similarity of Δf profiles between karyotype pairs. Our baseline reference (pairs with

substantially different karyotypes from different cell lines) had a Pearson coefficient not significantly above zero. Similarity was significantly higher for karyotypes drawn from the same inferred fitness landscape, which may reflect shared epistatic structure, but could also result from artefactual similarity introduced by using a single model: the Kriging-based fit encourages smoothness across nearby karyotypes, and shared input data can propagate systematic biases to all karyotypes within the same landscape. However, this effect was not limited to shared landscapes—karyotypes differing by fewer than three CNAs showed significantly more similar profiles even after controlling for landscape membership and irrespective of treatment status. This result suggests that modest karyotypic similarity is sufficient to produce convergent CNA fitness effects across otherwise distinct contexts, but that this relationship is fragile—diminishing rapidly as chromosomal distance increases. Whilst a recent analysis found significant correlations in CNA fitness across a large sample of cancers[8], it should be noted that CNAs in that study were all calculated relative to a euploid reference. Our results suggest such analyses could be refined by considering the impact of parental karyotype on CNA fitness effects.

Cells with similar karyotypes inhabiting the same fitness landscape peak can be conceptualized as a "quasispecies"[24]. A prediction of the quasispecies equation is that when mutation is sufficiently rapid, quasispecies will be unable to remain on narrow peaks in the fitness landscape, a concept known as the error threshold[24]. In an application

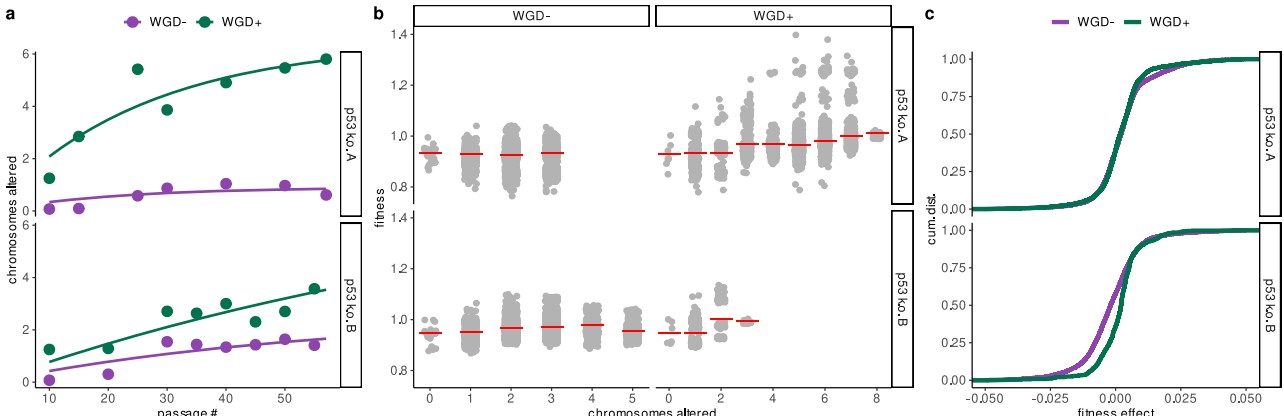

**Fig. 5 | Whole-genome doubling accelerates and reshapes aneuploidy evolution. a** Passaging trajectories for p53 k.o. A (top) and p53 k.o. B (bottom): number of chromosomes differing from the euploid state plotted by passage number, colored by WGD status (WGD⁻ = purple, WGD⁺ = green). Curves show fits from a non-linear saturation model accounting for additive effects of WGD status and trajectory on accumulation dynamics (see Supplementary Note 4.3). **b** ALFA-K–estimated relative fitness versus number of chromosomes altered: light-gray points are individual

fitness estimates ($n$ = 24,498 estimates derived from 2607 unique karyotypes pooled from the $n$ = 2 independent biological lineages); red ticks denote mean within each missegregation bin. **c** Empirical cumulative distributions of one-MS-step fitness effects for frequently observed karyotypes in each lineage, comparing WGD⁻ (purple) and WGD⁺ (green) subclones. Source data underlying these plots are available in the ALFA-K repository.

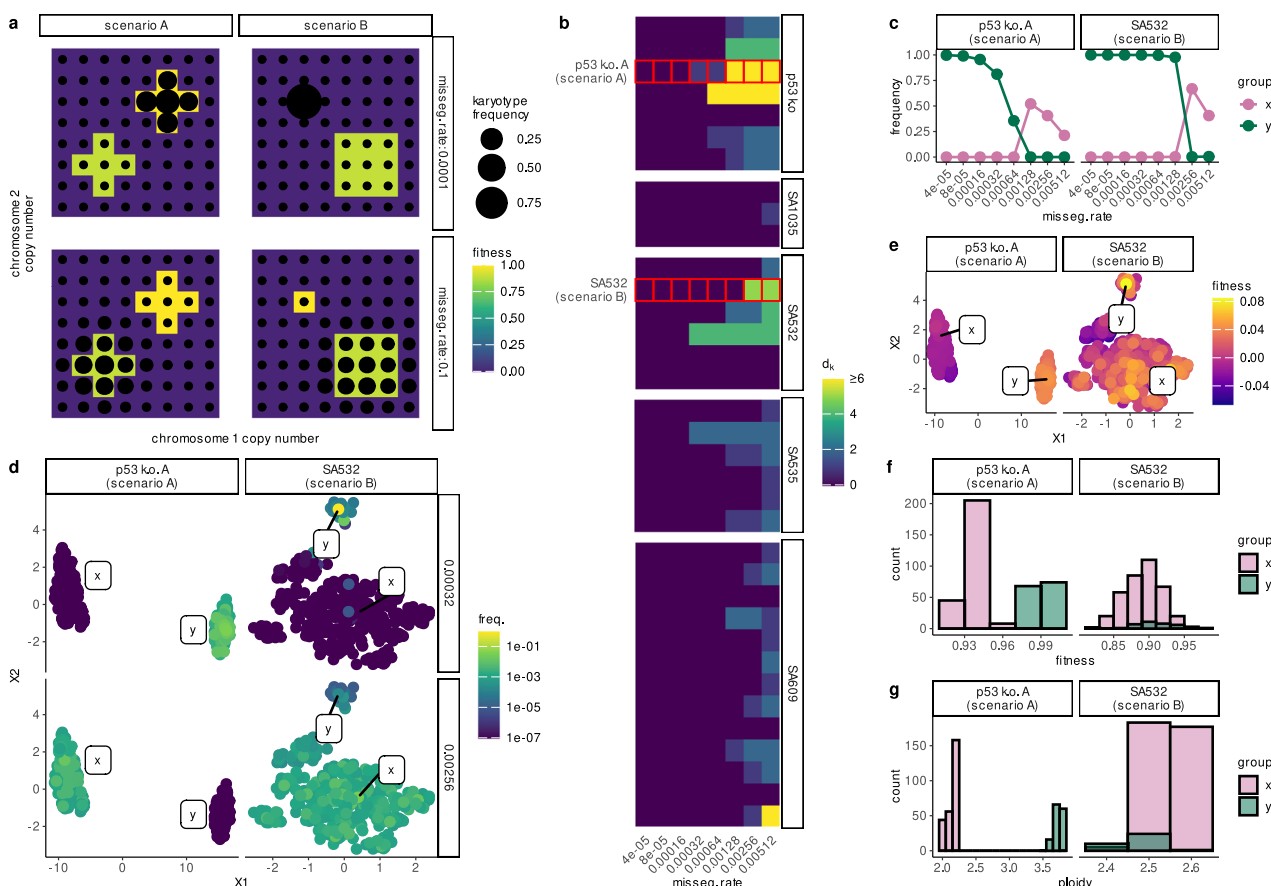

**Fig. 6 | Influence of missegregation rate on karyotype selection. a** Hypothetical fitness landscape for two scenarios (A: ploidy-driven; B: neighborhood-size–driven switches). **b** Heatmap illustrating screen results: fill color represents the Manhattan distance between the dominant karyotype at steady-state for the lowest missegregation rate compared to the dominant karyotype at the missegregation rate indicated. **c** Dominant group frequencies (x = high-rate; y = low-rate) in ABM

simulations for p53 k.o. A and SA532. **d** UMAP of all karyotypes colored by steady-state frequency at low (top) vs high (bottom) missegregation rates. **e** Same UMAPs colored by ALFA-K–estimated fitness. **f, g** Histograms of fitness (**f**) and ploidy (**g**) for the x/y groups identified in (**c**). Source data underlying these plots are available in the ALFA-K repository.

of ALFA-K, we investigated the impact of CIN on chromosomal evolution by increasing CIN rates until quasispecies were no longer sustainable on the highest fitness peaks of the landscape. This approach highlighted selection pressures against karyotype quasispecies residing on narrow fitness peaks or in high ploidy regions of the landscape. These findings emerged at missegregation rates of 5–20% per cell division, a range considered plausible for CIN cells[27]. Given that CIN is therapeutically modifiable, our results underscore the potential of using evolutionary principles to steer tumor evolution for long-term control[28]. The ploidy-dependent mechanism builds on prior models, which generally assume equal missegregation rates per chromosome, causing high-ploidy karyotypes to missegregate more frequently overall. While this assumption aligns with observations that high-ploidy cells missegregate more frequently[26,29], it warrants further scrutiny. Future models could benefit from integrating metrics such as interferon gamma as a measure of missegregation rate[25], as chromosome missegregations can trigger micronuclei formation, whose rupture activates the cyclic GMP-AMP synthase–stimulator of interferon genes (cGAS–STING) pathway and induces interferon-stimulated gene expression[30]. This could help discern whether a permissive fitness landscape or a propensity to missegregate underlies the relationship between WGD and aneuploidy.

One notable limitation of our study is the scarcity of longitudinal data available for training and validating ALFA-K. To mitigate this, we relied heavily on synthetic data and developed metrics to assess the accuracy of the inferred landscapes. However, the absence of independent biological replicates restricted our ability to fully evaluate the quality of our model predictions, as the inherent variability in the system was not well characterized. Future studies with identical biological replicates are essential to further validate ALFA-K[16]. Additionally, data from evolving cell populations inherently focuses on high-fitness regions, making estimation of the fitness impact of deleterious CNAs challenging. ALFA-K attempts to estimate fitness for unobserved karyotypes, using their absence to infer upper fitness limits. However, our estimates for the fitness impacts of deleterious CNAs are likely biased upwards and should be interpreted with caution. Unbiased screens that can accurately estimate the fitness impact of deleterious CNAs would significantly enhance our understanding[26]. A further limitation is that our current implementation treats each chromosome homogeneously, ignoring which allele is lost or gained. In cases where loss of a specific allele has distinct fitness consequences, allele-specific karyotypes would be more appropriate. ALFA-K could be applied directly to such data–simply by tracking each chromosome copy separately–but this would double the number of variables and likely necessitate additional longitudinal data for reliable interpolation.

We parameterized ALFA-K using a fixed missegregation rate consistent with literature averages[2,31,32], but chromosomal instability rates exhibit significant inter- and intra-tumor heterogeneity[25]. Because our model estimates the fitness of rare 'neighbor' karyotypes by accounting for mutational influx, the steepness of inferred landscape slopes is mathematically coupled to the assumed error rate. While our sensitivity analyses confirm that the fitness ranking of dominant clones is robust to this parameter, incorporating patient-specific empirical rate measurements will be essential for future studies aiming to rigorously decouple the evolutionary effects of high mutational supply from those of permissive fitness landscapes.

By quantifying Karyotype Fitness Landscapes (KFLs), the ALFA-K framework holds significant promise for forecasting long-term evolutionary dynamics in cancer. The application of ALFA-K to hematological malignancies offers a promising future direction for both understanding and treating these cancers. Hematological malignancies, of which an estimated 50% are aneuploid[2], are particularly suited for longitudinal sampling[33], providing the necessary data for accurate fitness landscape reconstruction. Metastatic colonization requires simultaneous adaptation across potentially hundreds to thousands of genes[34,35], a challenge inefficiently met by sequential point mutations alone. Chromosomal missegregation is a uniquely potent mechanism capable of supplying this complexity in a single leap, yet the multi-generational process of selection on this variation leaves a detectable evolutionary trail. By quantifying the steepness of local slopes and the width of neighboring peaks, ALFA-K converts those trails into in situ maps of evolutionary distance and direction. Similarly, long-term drug responses depend critically on how a tumor population traverses its fitness landscape. Therapeutic interventions that increase chromosomal missegregation rates can push populations beyond their error threshold, causing rapid extinction in rugged KFLs or enabling resistance emergence in smoother ones. In principle, the evolutionary path a clone must travel–from a low-fitness valley to a tissue-specific peak, or from a narrow, drug-sensitive summit to a broader, tolerant one–can be modeled using these inferred landscapes. Thus, ALFA-K charts the terrain that governs these adaptations, providing a quantitative basis to test hypotheses about their timelines without first needing to observe them. Because evolution across KFLs unfolds gradually–over tens to hundreds of cell generations[16,23]–ALFA-K inference can potentially forecast clinical timelines with actionable lead times, guiding treatment decisions aimed at preventing metastatic outgrowth or the evolution of drug resistance.

## Methods

### The ALFA-K method for inferring karyotype fitness

**Overview of the ALFA-K inference workflow.** We developed ALFA-K (Adaptive Local Fitness landscape for Aneuploid Karyotypes), a method to infer karyotype-specific fitness landscapes and forecast evolutionary trajectories from longitudinal karyotype count data. The core workflow involves three main inference stages (items 1–3) followed by validation (4) and forecasting (5):

1. **Frequent karyotype fitness estimation:** We identify frequently observed karyotypes and estimate their fitness using replicator dynamics.
2. **Single-step neighbor fitness estimation:** We extend fitness estimates to karyotypes that are one missegregation event away from the frequent karyotypes.
3. **Global landscape inference via kriging:** We use Gaussian process regression (Kriging) to interpolate fitness across the broader karyotype space.
4. **Internal consistency check:** We employ a cross-validation procedure to assess the reliability of the inferred landscape.
5. **Evolutionary forecasting:** We utilize the inferred fitness landscape within a simulation framework to predict future population dynamics.

**Fitness estimation for frequent karyotypes and their one-MS-step neighbors.** The fitness of frequently observed karyotypes (indexed by the set $S$) was modeled using the continuous-time replicator equation, which describes how the relative frequency $x_i$ of a karyotype $i$ changes over time under the influence of its fitness $f_i$ and the mean population fitness:

$$\frac{dx_i(t)}{dt} = x_i(t)\left[f_i - \sum_{j \in S} x_j(t)f_j\right]. \tag{1}$$

Fitness parameters were first initialized via Quadratic Programming (Supplementary Note 1.2.2) and then refined by maximizing the multinomial likelihood of the observed counts. The detailed mathematical derivations, along with procedures for growth offset correction and bootstrapping, are provided in Supplementary Note 1.2. These fitness estimates for frequent karyotypes were then extended to sparsely observed 'single-step neighbor' karyotypes by modeling the muta-

tional flux from their frequent parents:

$$\frac{dx_i}{dt} \approx \underbrace{\sum_{j \in S} P(\alpha_i | \alpha_j) f_j x_j(t)}_{\text{Flux from frequent parents}} + \underbrace{f_i x_i(t)}_{\text{Growth of neighbor } i} . \quad (2)$$

Here $P(\alpha_i | \alpha_j) = \text{Pr}(\textbf{one missegregation with per-chromosome rate } p$ **converts** $\alpha_j \rightarrow \alpha_i$). The neighbor-specific fitness $f_i$ was inferred by maximizing the likelihood in Eq. 10, which combines the binomial observation term with a normal prior on the fitness difference $\mathcal{N}(f_i - f_j | \mu_\delta, \sigma_\delta^2)$ for each parent $j$. The full derivation is provided in Supplementary Note 1.3.

**Global landscape interpolation.** A comprehensive fitness landscape was constructed using Gaussian process regression (Kriging) with a Matern kernel ($v = 1.5$), implemented using the `Krig` function in the R package `fields`. This process used the fitness estimates of frequent karyotypes and their one-MS-step neighbors as anchor points. The reliability of interpolated values was assessed via a bootstrap procedure detailed in Supplementary Note 1.4.1.

**Agent-based model for validation and forecasting**
We developed an agent-based model (ABM) to generate synthetic data for validation and to perform forward predictions. In the ABM, cells divide at a karyotype-specific rate (fitness), with a defined per-chromosome missegregation probability ($p$) generating variation. The model was run in two modes: 1) using a known Gaussian-random-field (GRF) fitness landscape to generate synthetic data for benchmarking ALFA-K, and 2) using a look-up table (LUT) of fitness values inferred by ALFA-K to forecast future population dynamics. To emulate cell culture experiments, population size was controlled via simulated serial passaging at a defined maximum capacity ($N_{\max}$). Full implementation details and simulation parameters are provided in Supplementary Note 2 and Supplementary Table 1.

**Method validation**
**Validation on synthetic data.** We ran ABM simulations for 300 days on GRF landscapes with varying complexity ($\lambda$). Key parameters are listed in Supplementary Table 1. For testing ALFA-K, we sampled data (simulating experimental measurements) from these simulations. We extracted longitudinal count data for 2, 4, or 8 consecutive passages ending around day 120 of the simulation. This timeframe usually captured populations during active adaptation (Fig. 1e), allowing us to test ALFA-K's ability to infer ongoing dynamics and predict future evolution. We compared the inferred fitness landscapes against the known ground truth to assess accuracy. The validation showed that inference accuracy, primarily measured by Spearman's correlation, improved with smoother landscapes and an increased number of sampled time points. The validation procedure and full performance results are detailed in Supplementary Note 3, Supplementary Figs. 1, 2.

**Internal consistency and forecasting accuracy.** We developed a leave-one-out cross-validation procedure and a resulting metric, CV score, as a key diagnostic for the reliability of the inferred landscape (Supplementary Note 1.5). A positive CV score was strongly correlated with high accuracy against the ground-truth fitness landscape. We then tested the ability of the inferred landscapes to predict future evolution, and found that forecasts from landscapes with a positive CV score were significantly more accurate than random and consistently outperformed simple no-evolution baseline models. Full forecasting methodology and results are available in Supplementary Note 3.4 and Figs. 4, 5.

**Application to experimental data**
The ALFA-K method was applied to a longitudinal single-cell DNA sequencing dataset[23], which characterized karyotype evolution in immortalized human mammary epithelial cell lines (184-hTERT) and four PDX models. For each cell, we generated a 22-element integer karyotype vector by taking the modal integer copy number across all genomic bins assigned to each autosome (see Supplementary Note 4.1).

**Statistical analysis**
**Analysis of directional consistency.** To quantify the directional accuracy of evolutionary forecasts in karyotype change, we computed angles between paired karyotype distributions, summarizing each experimental unit by the median of its angles. Each unit corresponded to a serially passaged population derived from one ancestral sample, or in the unrelated control, a pair of such populations from distinct models. The global test statistic $T$ was defined as the median of these per-unit medians. To evaluate whether $T$ was smaller than expected under random directional change in 22-dimensional space, we performed a Monte Carlo test using the theoretical distribution of angles between random unit vectors. In each of 10,000 iterations, synthetic angles were drawn to match the observed sample sizes per unit, and $T^*$ was recomputed. The $p$-value was defined as the proportion of $T^*$ values less than or equal to the observed $T$ (one-sided for model forecasts and sister passages, two-sided for unrelated controls).

A comprehensive suite of additional metrics was used to evaluate performance; full definitions are provided in Supplementary Note 3.1.

**Prediction of novel karyotype emergence.** For each lineage, the latest passage ($S_0$) was used to predict outcomes in the next ($S_t$). A karyotype was considered *novel* if it was absent from $S_0$ but present in $S_t$. Karyotypes were encoded as 22-digit copy-number vectors; Manhattan distance to every $S_0$ karyotype yielded covariates $d_1 – d_5$, representing the fraction of the $S_0$ population one to five missegregations away. ALFA-K fitness values (**f**) were computed from landscapes trained only on data up to $S_0$, ensuring no look-ahead. We retained fits with a CV score > 0 and at least three passages. When multiple fits terminated at the same passage, we retained only the one with the highest CV score, so that each terminal passage defined a single, independent test instance. For each retained fit, we modeled the probability of emergence with a binomial logistic regression: $\text{logit}\,P(\text{novel}) = \beta_0 + \beta_f f + \sum_{i=1}^{5} \beta_i d_i$. Significance of each predictor was assessed via Wald $z$-tests at $P < 0.05$ (R 4.3, `stats::glm`).

**Analysis of karyotype background influence on fitness landscapes.** To investigate how the properties of fitness landscapes were shaped by biological context, we first addressed the statistical dependencies caused by overlapping evolutionary trajectories in our data. We implemented a bootstrap framework where, for each of hundreds of replicates, a random set of non-overlapping lineages was algorithmically selected. This method ensured that each replicate of our analysis was based on statistically independent data, allowing for a robust assessment (see Supplementary Note 4.2 for details).

We then used Generalized Linear Mixed Models (GLMMs; `glmmTMB` R package) to analyze the distributions of potential fitness effects ($\Delta f$) on these trajectories. We fitted separate models to test how context (PDX vs. in vitro) and drug treatment influenced the magnitude and the variance of fitness effects. These models included nested random effects for the cell line, specific trajectory, and focal karyotype to account for the data structure.

Finally, to assess landscape smoothness, we used a Linear Mixed Model (LMM; `lme4` R package) to model how the similarity between the fitness vectors of two karyotypes (Fisher-z transformed Pearson correlation) decayed as a function of the karyotypic distance between them. The full mathematical specifications for the trajectory selection algorithm and all mixed-effects models are provided in the Supplementary Note 4.2.

**Statistical modeling of whole-genome doubling (WGD) effects**. To investigate the impact of WGD, karyotypes were first classified as WGD$^+$ if their modal chromosome copy number was $\geq 3$. We then modeled the dynamics of aneuploidy accumulation over time using non-linear least squares (`nls` function in R). The average number of altered chromosomes per cell was fitted to an exponential saturation curve, where the rate and asymptote parameters were modeled with fixed effects for WGD status and experimental trajectory. The full specification for this non-linear model is detailed in Supplementary Note 4.3.

To directly compare the fitness consequences of mutations, the distributions of single-chromosome gain/loss fitness effects ($\Delta f$) for WGD$^+$ versus WGD$^-$ states were compared using a permutation Kolmogorov-Smirnov (KS) test. To avoid pseudo-replication, mean effects were calculated per WGD status within each landscape fit before comparison. Significance was then assessed against $10^4$ permutations of the WGD labels among the landscape fits.

### Reporting summary
Further information on research design is available in the Nature Portfolio Reporting Summary linked to this article.

### Data availability
The large dataset containing Agent-Based Model (ABM) simulation data used for synthetic benchmarking is available via Zenodo[36] https://doi.org/10.5281/zenodo.17726562. The raw experimental data originating from Salehi et al.[23], as well as data required to reproduce the specific analyses presented in this work, are available in the project repository[37]: https://github.com/Richard-Beck/ALFA-K; https://zenodo.org/doi/10.5281/zenodo.17753550.

### Code availability
The core ALFA-K method is implemented as a lightweight R package (*alfakR*), which is available[38] at https://github.com/Richard-Beck/alfakR. Source code and scripts required to generate the results and figures presented in this paper are available[37]: https://github.com/Richard-Beck/ALFA-K.

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

## Acknowledgments

We are grateful to the members of the Integrated Mathematical Oncology (IMO) department at Moffitt for their valuable feedback on this manuscript. The insightful comments and suggestions provided during numerous department meetings were instrumental in strengthening the final work. We also thank Sohrab Shah and Sohrab Salehi for sharing the scDNA-seq–derived copy number calls, along with detailed information on their generation. This work was supported by the NCI grants 1R37CA266727-01A1 (N.A), 1R21CA269415-01A1 (N.A) and 1R03CA259873-01A1 (N.A). The funders had no role in study design, data collection and analysis, decision to publish, or preparation of the manuscript.

## Author contributions

N.A. and R.J.B. conceived the study; R.J.B. and N.A. developed the mathematical methodology; R.J.B. implemented the ALFA-K software and performed the formal analysis; T.L. performed software and methodological validation; N.A. acquired funding and supervised the project; R.J.B. and N.A. wrote and edited the manuscript.

## Competing interests

The authors declare no competing interests.
