## [Transparent Peer Review file · Nature Communications]

ALFA-K: Local Adaptive Mapping of Karyotype Fitness Landscapes

Corresponding Author: Dr Noemi Andor

Version 0:

Reviewer comments:

Reviewer #1

(Remarks to the Author)

Beck and Andor develop a new method called ALPHA-K, which aims to estimate the fitness values of different karyotypes, from longitudinal single cell sequencing data. They validate their model using synthetic data obtained from an agent-based model, before evaluating their method on previously published single-cell breast cancer data. They apply their method for a variety of purposes, such as predicting future karyotypes, evaluating the role of cisplatin and WGD on fitness landscapes and examining the role of varying missegregation rates on the distribution of karyotypes. This work is interesting and there is a need for the effect of specific copy number changes on fitness to be better studied. However, I have a number of concerns that I would like to be addressed, particularly concerning the validation of the model.

Major Points

- The authors appear to assume from equation (1) that all common karyotype populations are present in tumor from initiation and are growing at a fixed exponential rate. This is a strong assumption and is insufficiently justified. It doesn't account for progressive evolution from diploid and that there may be late emerging common states with a complex hard-to-reach karyotype and high fitness. Furthermore, I would like to see the assumptions and approximations that ALPHA-K makes in its modelling to be more clearly stated and justified, with simulations if appropriate, in the text.
- The cross-validation procedures outlined in the method are insufficient. They serve to demonstrate that the fitness values obtained from step 1 of the method are concordant with those produced from steps 2 and 3, but not that the method itself overall is accurate. The statement that the fitness estimates produced from step 1 are 'relatively robust' is not justified. It is unclear to me why the true fitness values from the agent-based model did not appear to be directly used in the validation. One suggestion would be to employ an ordinal regression method between the true fitness values from the Gaussian random fields and the values inferred from all three steps of the pipeline.
- ALPHA-K appears to produce output with a very poor fit between the fitness estimates from step 1 and step 2+3 ($R^2 < 0$), commonly when analyzing low numbers of longitudinal samples or a complex fitness landscape. This needs to be explored more, such that the reader can understand what is occurring under these circumstances, and for which data the estimates from ALPHA-K is robust.
- As currently applied, the angle metric is not a sufficient validation for future karyotype predictions. I don't believe that random draws from hyper-sphere is a useful baseline. It is not the case that obtaining an angle of less than 90 degrees 'indicates better agreement than random chance'. From figure S4, approximately 50% of the pairs of sampled hypersphere points have a smaller angle than this.

A more robust null distribution is required to test the predictive power of ALPHA-K. The comparison between the angles obtained from ALFA-K/ABM populations and branching lineages in figures 3C&D is good but it would be useful to have a figure where they are directly compared. One suggestion would be to compare the distance between predicted and true karyotypes to the distance between the karyotypes from unrelated samples.

- The theoretical models of the effect of missegregation rate on karyotype are interesting. However, it is unclear with what frequency the predicted effects are observed. 'For a subset of fitted landscapes' – under what kinds of landscapes do these results hold?

Minor Points

- Lines 65+304 - I would avoid the use of the term focal as this typically is used to refer to copy number events much smaller than the size of an entire chromosome, which is what I understand to be the scale that ALPHA-K operates at. The scale for ALPHA-K should be explicitly clarified in the text.

- Is the time t a quantitative measure of the time between each sampling or an index variable for the samples? It's use both in equation one and as a summing index in the log-likelihood suggest different roles.

- Line 86 - The likelihood of obtaining the karyotypes $y_{\{t\}}$ given $u_{\{t\}}$ is not equal or proportional to the product of the binomial marginal distributions for each $y_{\{it\}}$ as they are not independent. It should be a multinomial likelihood function, although this should still have the same maxima in $u_{\{t\}}$ as the product of binomials. This should be clarified or corrected. Similarly, could the authors write out an explicit definition for the likelihood, including the parameters it is conditioned on.

- It would be worth considering fixing karyotypes with the complete loss of any chromosome as having a fitness value that makes them effectively unsurvivable.

- More details on the terms $P(\alpha_{\{i\}} | \alpha_{\{j\}})$ and $Q(f_{\{i\}} - f_{\{j\}})$ used in the nearest neighbor fitness calculation should be given in the main text.

- Are the fitness estimates f used in the binomial logistic regression in figures 4E&F estimated from all longitudinal samples or just the samples before the novel karyotype emerged? If the latter, it would be good to observe a comparison with the former.

- Line 253 – I wouldn't say that 'CNAPs across the two untreated lineages exhibited much more evidence of correlation than CNAPS compared between either untreated lineage and the cisplatin treated lineage'. In both 5F and 5G, the mean Pearson correlation is very close to zero for each Manhattan distribution and the distribution of the individual comparisons is relatively broad.

- Why does the maximum number of chromosomes altered vary between cell line and WGD population in figure 6B? Is this due to a limit on how far from the observed population ALPHA-K can estimate karyotype fitness? This should be clarified.

- It is worth mentioning the limitation of only considering total copy number in the discussion. For example, in the context of an inactivating tumor suppressor gene mutation, a $1+1 \rightarrow 1+0$ event where the unaffected allele is lost will have a very different fitness effect to a $2+0 \rightarrow 1+0$ loss.

Typos

- Line 29 - Should "gene dosage" be "gene dosage"
- Equation 2 – $v_{\{i\}t}$ should be $v_{\{it\}}$.
- Line 226 – 'significant'

(Remarks on code availability)

I have checked the repository and found that the code for ALPHA-K and the agent-based modelling is present. There is a minimal README but probably not enough to run either of the tools easily. It would be good to document the format for the input files needed to run ALPHA-K and how to interpret the output.

Reviewer #2

(Remarks to the Author)

Summary:

This paper reports on a novel method to estimate karyotype fitness scores from single cell copy-number alteration data. They use a three step method to obtain fitness scores in the neighborhood of frequent (most fit) karyotypes and include a leave-one-out method to assess quality of their model fit. They validate their method with synthetic data (producing a ground truth) from an agent-based model operating on a Gaussian random field landscape. The method is then used to fit a series of publicly available data sets including cell line and patient derived xenograft data. Using the estimated fitness scores, the authors find several insights suggested by their analysis including effects of whole genome doubling and cisplatin treatment on fitness landscapes and the role of chromosome missegregation rate on tumor cell population "quasispeciation" and evolution.

The authors make the need for this method clear both with reasonable literature search and useful applications. There are small errors throughout the manuscript (see line by line comments) and some sentences might benefit from reworking to improve readability. The visualizations and figures do an appropriate job of conveying the author's method and findings. See line by line comments for additional comments.

Regarding flow of the paper, I suggest the authors consider a slightly different ordering. Perhaps the results should come first as many of the methods become clearer through example (at least for this reviewer). Methods could then be presented at the end of the paper with additional details in the SM. If this does not fit the authors' style, I strongly suggest moving some aspects of the results and diagrams/aspects of diagrams from results into methods - to make methods more understandable and concrete. At a minimum, referring to diagrams within the results would likely aid readers (especially Figure 1). If the authors produced a more detailed methods section (for the SM likely) they also could include an expanded visualization and discussion of the inference and prediction matrices, the definition of nearest neighbors and other important concepts that underlie their interesting methodology.

Finally, as noted in more detail below, the paper would benefit from additional detail at focused points throughout. I especially want to highlight adding additional details of simulations run in the methods section, including a description of the biological data used in the results, and more strongly tying the results into the discussion in particular to aid in highlighting how the results from the novel method may also be novel, hypothesis generating results for the biology.

Line by line comments:

11 - "not-yet seen in the data" is confusing to me. This sentence may benefit from reworking. Does it mean "enabling the prediction of emergent karyotypes from input data"?

29 - Inconsistent use of "mis-segregations" versus "missegregations" (see line 91 for example).

29 - is "gene dosage" a typo?

32 - missing space

38-40 - Sentence reads in a confusing fashion. Maybe "do not independently..." instead of "are not independently"?

41 - Maybe helpful for reader to call out the difference between chromosomal instability and aneuploidy and/or various kinds of chromosomal instability and the process(es) by which it causes aneuploidy

Literature review - Authors maybe interested in this article - which takes a different approach to exploring CIN:
<https://elifesciences.org/articles/69799>

64 - Does ALFA-K stand for something? If it does, would you consider laying out what it stands for?

78 - Typo: LaTeX leading double quote needs used: ``

81 - Likely typo: New paragraph was used mid-sentence.

~81- Could the authors define v_0 ? Apologies if I simply missed the definition.

77-87 - This reviewer is somewhat lost here. Would the authors consider laying out the story of the equations with more detail given how critical this is their method? If appropriate, detail could be added to the supplement, allowing the interested reader to go there for more information/exposition.

Line 79 - The authors define a matrix U but never use it. Is it \hat{U} ?

Line 85 - It might be clearer to say "Defining S to be the set of indices belonging to frequent karyotypes"?

Line 89 - What does von Neuman (I believe the "v" isn't capitalized) neighbor mean in the context of this non-spatial model? Could the authors please explicitly call out what they mean? I am guessing von Neuman karyotypes are karyotypes that differ by one count in one dimension in a 22 member vector continuing counts of each chromosome. Is that correct?

Line 93 - For clarity, please consider calling out Equation 4 as "Equation 4" or something similar rather than simply "4".

Line 91 - Likely typo: New paragraph was used mid-sentence.

Line 100 - Please capitalize "Gaussian".

Line 101 - As "fields" is a common noun, consider calling out the package name with quotes or in ``code`` type font.

101 - "von Neuman".

111 - "data set" is used rather than "dataset". The paper typically uses "dataset". Please consider updating line 111.

113 - Would the authors define R^2 either in line or referring the reader to where its defined in the text?

116 - In this section, could the authors lay out explicitly how the ABM is used in the inference process? It appears to be used

in the inference process later in the paper, not only for generating synthetic data for testing.

120 - To aid readers, I suggest the authors add a brief overview of the methods and corresponding per-chromosome missegregation rates in their supplementary material rather than referring out for these critical methods and parameters?

126 - Could the authors please cite literature or otherwise justify the parameters used in their simulations? As appropriate, readers could be referred to SM.

Line 136 - Equation 6 - would the authors consider increasing the size of the parentheses for better presentation (using “left(“ and “right”) for example)? Also, please use Roman type font for “sin”.

Line 139 - Could the author explain/expand on the α_{ik} and α_{ij} as vectors given that in index k seems to represent only a single entry in the karyotype space K ? To the reviewer, at least it seems like a_{ik} might represent a single dimension in karyotype i rather than the whole 22 dimensional karyotype vector.

Line 144 - Inline equation is missing additional index i for variable n .

Line 146 - Would the authors consider specifying N_i in words? I believe its the total number of chromosomes in karyotype i ?

148 - New paragraph mid-sentence.

Line 150 - This appears to be a great summary of the methods. As noted in my summary comments, bringing this summary up earlier in the manuscript would benefit readers as well as including the paragraphs below. In some sense, the initial “results” included may be considered “methods” in the sense that these first results illustrate and validate the method. Figure 3 could be reconceived as the first result - when the method meets the non-synthetic data.

Figure 1: B - Could the authors include the value of λ for this example? D: If each karyotype can only change by one missegregation, how do agents arise that differ by more than one chromosome? Are there transient, low frequency populations not included?

177 - Typo: `` needed.

172 - Would the authors please explain/describe which simulations/computational experiments they performed? Like - the values of λ used, the number of stochastic replicates per experimental condition, etc.

176 - Could the authors please explain what specifically they mean by number of longitudinal samples? I assume its the number of stochastic replicates?

183 - Would be helpful to explain angle metric inline.

185 - Could the authors provide an interpretation of a score and/or a landscape that produces a R^2 score that is positive versus 0 versus negative? I assume it is simply that the algorithm fails to characterize these landscapes (at least with the given input stochastic replicates)?

Figure 2: E: I suggest the authors consider increasing the α value or adding a darker border to the individual sample points. The points in yellow end up presented as rather fuzzy, making it difficult to see the boundaries of the distribution. Also, what is meant by the rows frequent, distance 1 and distance 2? I do not see those designations called out.

194 - Could the authors please provide more background for this data? What is SA532? What are the relevant experimental conditions that generated these data? How many lineages are in the data set and what are they from?

197-198 - Something is off with the sentence beginning “Thus e. g. ...”. Consider starting with “For example, in SA532 ... “?

200-201 - For clarity, please consider adding commas after “found” and “results”.

206 - Missing space in text.

Figure 3: C and D captions: Could the authors call out ECDF in the caption so that the caption matches the diagram?

216 - 217 - This appears to be a run-on sentence.

Figure 4: Caption: A) what is clonal interference? Could the authors explain what that means in this context and how its related to the experiment? Overall, using the caption and/or text could the authors relate/connect 4A back to the text?

There is a typo in the second line.

231 - PDX datasets have not yet been discussed at this point. Could the authors discuss what data sets they are working as well as define the abbreviation?

234 - Which conditions?

243 - There is an exclamation point rather than an expected symbol.

254 - What quality control? Is this described in the manuscript?

Figure 5: Pearson needs capitalized. Manhattan needs capitalized.

290 - "dependent" is misspelled.

340 - 346 - Could the authors tie this narrative back to a specific set of their results?

341 - Typo: `` needed.

359 - Could the authors expand on what they mean by interferon gamma as a metric of missegregation? In what context is interferon gamma a metric of this?

368 - Typo: estimated

(Remarks on code availability)

Code:

Well organized repository with extensive READMEs and documentation of figures in paper
Code downloads and compiles.

Generates warning:

In file included from ./ABM/main.cpp:17:

```
./ABM/setup.h:6:26: warning: implicit conversion from 'long' to 'int' changes value from 10000000000 to 1410065408 [-Wconstant-conversion]
```

```
int pop_write_freq = 10000000000;
```

```
~ ^~~~~~
```

```
1 warning generated.
```

Explored one example.

Running example 1:

https://github.com/Richard-Beck/ALFA-K/tree/main/examples/example_1

Note - I don't have the package `lhs`

```
> source("utils/ALFA-K.R")
```

```
Error in library(lhs) : there is no package called 'lhs'
```

Produces similar output to example in README, including plot.

Runs until Process output from ABM simulation

Get error that proc_sim isn't available. Similar error for alfak function

Demonstrated code compiles and at least a good portion runs. Note I didn't try to troubleshoot - just running as is. Additional investigation showed that as expected, the functions proc_sim and alfak are in the R - just for some reason not pulling up on my machine. Perhaps something is missing in README in this section?

Thank you for making your code immediately available and organized.

Data repository:

Examined but did not download due to repository size. Appears as expected. Thank you for making your data easily available.

Version 1:

Reviewer comments:

Reviewer #2

(Remarks to the Author)

See attached file.

(Remarks on code availability)

Reviewed two repositories. Spot checked that it contains the expected languages (C++ and R) and sensible syntax.

Appears to be complete. Repository appears to be thoroughly annotated with READMEs as well as scripts to run both the complete results of the paper as well as a separate example, positioned as a locally installed R package.

Reviewer #3

(Remarks to the Author)

This study demonstrates a computational framework (ALFA-K) to determine the fitness landscape for varying karyotypes based on time-series scWGS data. This framework is thoughtfully designed, and its application to data from the Shah lab yields valuable insights into the context-specific effects of aneuploidy in cancer. Nevertheless, I suggest several points for improvement:

Assumption on pre-existing karyotypes:

The method assumes that all frequent karyotypes are pre-existing. This raises questions about the simulation setup. Specifically, how the initial fractions of these karyotypes are determined and whether they can be accurately estimated from synthetic data. It would be helpful to report the accuracy of initial fraction estimation derived from joint likelihood optimization.

Effect of missegregation rate on fitness inference:

It would be useful to assess whether increasing the missegregation probability in the agent-based model affects the accuracy of fitness inference. In particular, transitions between frequent karyotypes could potentially obscure or confound the inferred fitness landscape.

Details on growth offset correction:

ALFA-K models the frequency of karyotypes by an ODE, assuming the population follows the exponential growth model. To infer the fitness of frequent karyotypes, the author conducts a two-step optimization process: first solving a QP, followed by maximum likelihood estimation. This approach is appropriate for fraction data. However, in the subsection "Growth Offset Correction for Passaging Experiments", the manuscript states that predicted fitness values can be shifted to obtain an exponential growth rates. However, this shift is not obvious, and it might be beneficial to provide more details here to clarify.

Minor comments:

The equation S3 has a typo in objective function, where r seems undefined.

For notation in model definition, it might be helpful to define the initial frequencies as $x_{i,0}$ to make it consistent with definition of other $x_{i,t}$.

In page 8 line 190, a typo (Fig 4E -> Fig 4F)

(Remarks on code availability)

Reviewer #4

(Remarks to the Author)

(Remarks on code availability)

The code is executable.

5 Response to reviewers

5.1 Reviewer 1 Major Point 1

The authors appear to assume from equation (1) that all common karyotype populations are present in tumor from initiation and are growing at a fixed exponential rate. This is a strong assumption and is insufficiently justified. It doesn't account for progressive evolution from diploid and that there may be late emerging common states with a complex hard-to-reach karyotype and high fitness. Furthermore, I would like to see the assumptions and approximations that ALPHA-K makes in its modelling to be more clearly stated and justified, with simulations if appropriate, in the text. *We thank the reviewer for raising this important point. In the revised Methods section, we now explicitly state the assumption that frequent karyotypes are modeled as growing exponentially from the beginning of the sampling window, without explicitly modeling their emergence via missegregation. This approximation avoids the need to reconstruct complex mutation histories or infer upstream parent clones — which is often underdetermined, as many frequent karyotypes can arise through multiple missegregation paths. By contrast, for rare one-step neighbors of frequent clones, we do estimate a parental birth time, since the mutational path and timing are simpler and locally defined. We make this approximation because it allows each frequent clone's fitness to be estimated directly from its observed frequency trajectory, without constraining or biasing the result. The initial abundance of each clone is fit jointly with its growth rate and can be arbitrarily small. In this way, the model accommodates late-arising clones while maintaining a tractable and transparent inference procedure. In addition, we have expanded the Methods section (Supplementary Section 1) to describe other assumptions relevant to each major step of the ALFA-K pipeline (Frequent karyotypes, one-MS-step neighbors, All Other Karyotypes (Kriging), Cross Validation Procedure), making the modeling framework more transparent overall.*

5.2 Reviewer 1 Major Point 2

The cross-validation procedures outlined in the method are insufficient. They serve to demonstrate that the fitness values obtained from step 1 of the method are concordant with those produced from steps 2 and 3, but not that the method itself overall is accurate. The statement that the fitness estimates produced from step 1 are 'relatively robust' is not justified. It is unclear to me why the true fitness values from the agent-based model did not appear to be directly used in the validation. One suggestion would be to employ an ordinal regression method between the true fitness values from the Gaussian random fields and the values inferred from all three steps of the pipeline. *We thank the reviewer for this important point. In the revised manuscript we have added a dedicated "Accuracy Metrics" subsection in the Supplementary Information and now benchmark ALFA-K's fitness estimates directly against the known ground truth from the agent-based simulations. In section 3.2 we now report the Spearman rank correlation (ρ), Pearson correlation (r), and a centered R-squared value (R^2) between the inferred fitness scores and the true fitness assigned to each karyotype. These metrics are reported across 400 synthetic datasets spanning a wide range of landscape complexities and sampling depths. We believe these results fulfill the request for external validation beyond internal consistency checks. We have also removed the phrase "relatively robust" from the Methods text, in line with the reviewer's suggestion.*

5.3 Reviewer 1 Major Point 3

ALPHA-K appears to produce output with a very poor fit between the fitness estimates from step 1 and step 2+3 ($R^2 < 0$), commonly when analyzing low numbers of longitudinal samples or a complex fitness landscape. This needs to be explored more, such that the reader can understand what is occurring under these circumstances, and for which data the

1065 **estimates from ALPHA-K is robust.** *We have expanded the manuscript (Supplementary Section 3.2)*
1066 *to systematically analyze cases where ALFA-K fits fail. In the revised Figure S2, we trace low R-squared*
1067 *values to a specific mechanism: when there are too few longitudinal samples or the “frequent” karyotype*
1068 *threshold is set too low, stochastic noise can cause incorrect ordering of fitness among a small number of*
1069 *abundant clones. These errors then propagate through the subsequent steps of the pipeline. This diagnostic*
1070 *analysis helps clarify the conditions under which ALFA-K is reliable. Figure S2F specifically shows how*
1071 *poor fits (negative global R^2) often arise from poor initial fits to frequent karyotypes (negative R_f^2) or having*
1072 *too few frequent karyotypes.*

1073 **5.4 Reviewer 1 Major Point 4:**

1074 **As currently applied, the angle metric is not a sufficient validation for future karyotype**
1075 **predictions. I don’t believe that random draws from hyper-sphere is a useful baseline. It is**
1076 **not the case that obtaining an angle of less than 90 degrees ‘indicates better agreement than**
1077 **random chance’. From figure S4, approximately 50% of the pairs of sampled hypersphere**
1078 **points have a smaller angle than this. A more robust null distribution is required to test the**
1079 **predictive power of ALPHA-K.** *We agree with the reviewer that the hypersphere null model was not*
1080 *sufficiently rigorous. In the revised manuscript, we now use the exact probability density of angles between*
1081 *random unit vectors in \mathbb{R}^{22} as the null distribution (Eq. S17). This density is proportional to $\sin(\theta)^{20}$,*
1082 *and the cumulative distribution is plotted in the new Figure 2. Observed angles are now compared against*
1083 *this exact null to compute p-values. We have also strengthened the evaluation framework by introducing*
1084 *an independent “no-evolution” baseline: for each lineage, we compare ALFA-K’s prediction to the outcome*
1085 *expected if no further karyotype evolution occurred (i.e. at t_0 , the population has already reached the peak*
1086 *of the fitness landscape; Figure 2E-H, Supplementary Section 3.1).*

1087 **5.5 Reviewer 1 Major Point 5**

1088 **The comparison between the angles obtained from ALFA-K/ABM populations and branching**
1089 **lineages in figures 3C and D is good but it would be useful to have a figure where they are**
1090 **directly compared. One suggestion would be to compare the distance between predicted and**
1091 **true karyotypes to the distance between the karyotypes from unrelated samples.**

1092 *We reorganized Figure 2 (formerly Figure 3) so that panel C now compares ALFA-K forecasts (blue) with*
1093 *sister passages (green) and unrelated lineages (red); the plot shows that our directional predictions diverge*
1094 *to the same extent as replicate cultures, and both are markedly more accurate than the null baseline or the*
1095 *unrelated lineages. We also added panels to figure 2 to benchmark against a static “no-evolution” baseline*
1096 *and to compare forecast performance with the same metric applied to sister passages. These changes should*
1097 *help better evaluate the predictive ability of our pipeline.*

1098 **5.6 Reviewer 1 Major Point 6**

1099 **The theoretical models of the effect of missegregation rate on karyotype are interesting.**
1100 **However, it is unclear with what frequency the predicted effects are observed. ‘For a subset**
1101 **of fitted landscapes’ – under what kinds of landscapes do these results hold? *We clarified this***
1102 *in the results section corresponding to Fig. 6B. We state: “As an initial screen we searched for landscapes*
1103 *in which the most frequent karyotype differed across missegregation rates - 28 out of 35 tested sublineages*
1104 *met this criteria.”. We then explored two examples in more depth. The analysis investigates landscapes*
1105 *derived directly from the experimental data (SA532 and p53 k.o. A), showing that these rate-dependent*
1106 *switches can occur under conditions relevant to the biological systems studied.*

5.7 Reviewer 1 Minor Point 1

Lines 65+304 - I would avoid the use of the term focal as this typically is used to refer to copy number events much smaller than the size of an entire chromosome, which is what I understand to be the scale that ALPHA-K operates at. The scale for ALPHA-K should be explicitly clarified in the text. *We agree and have replaced "focal regions" with "local regions" in the abstract and discussion. The Introduction clarifies that ALFA-K operates at the whole-chromosome level: "Losses and gains of entire chromosomes or large sections thereof, known as aneuploidy..." and "This method utilizes longitudinal single-cell karyotype data...to estimate the fitness of thousands of karyotypes proximal to the observed data in karyotype space".*

5.8 Reviewer 1 Minor Point 2:

Is the time t a quantitative measure of the time between each sampling or an index variable for the samples? It's use both in equation one and as a summing index in the log-likelihood suggest different roles. *Time t is a quantitative measure (e.g., days or passages). The equations in the Methods (Supplementary Section 1) use t consistently as continuous time in the differential equations and as discrete sampling time points when calculating the likelihood based on observed counts y_{it} . We have clarified the notation where necessary.*

5.9 Reviewer 1 Minor Point 3

Line 86 - The likelihood of obtaining the karyotypes y_t given u_t is not equal or proportional to the product of the binomial marginal distributions for each y_{it} as they are not independent. It should be a multinomial likelihood function, although this should still have the same maxima in u_t as the product of binomials. This should be clarified or corrected. Similarly, could the authors write out an explicit definition for the likelihood, including the parameters it is conditioned on. *We thank the reviewer for pointing this out. We agree that the observed karyotype counts are not independent and that a multinomial likelihood better captures their dependency structure. In the revised manuscript (Supplementary Section 1.2), we have corrected the description and equation (now explicitly stating the multinomial likelihood) to reflect this. While the maximization procedure remains unchanged, this formulation more accurately reflects the probabilistic structure of our data. Additionally, we have clarified that the likelihood is conditioned on the total observed cell count N_t at each time point t . For the neighbours step, we use a Binomial likelihood as stated.*

5.9.1 Reviewer 1 Minor Point 4

It would be worth considering fixing karyotypes with the complete loss of any chromosome as having a fitness value that makes them effectively unsurvivable. *We agree that karyotypes missing an entire chromosome are likely non-viable. In fact, we already enforce this constraint in our agent-based model simulations, where such karyotypes are removed from the population. We now clarify this in the Methods section (Supplementary Section 2) of the manuscript. However, we do not apply a hard zero-fitness cutoff in the ALFA-K inference pipeline, specifically in the kriging step. This is because the kriging model assumes smooth variation in fitness across karyotype space. Imposing a discontinuous fitness boundary would violate this assumption and could distort interpolation in nearby regions. Instead any karyotypes with zero copy number on any chromosome are filtered out before the kriging step, a fact which we clarify in our revised manuscript (Supplementary Section 1.4).*

1147 5.10 Reviewer 1 Minor Point 5

1148 **More details on the terms $P(\alpha_i|\alpha_j)$ and $Q(f_i-f_j)$ used in the nearest neighbor fitness calculation**
1149 **should be given in the main text.** *We agree with the reviewer that these terms required clearer*
1150 *definitions. In the updated “Nearest Neighbours” Supplementary Section 1.3 we now:*

- 1151 *1. give an explicit closed-form equation for $P(\alpha_i|\alpha_j)$ as the per-division probability that a single misseg-*
1152 *regation in parent karyotype α_i produces α_j (Eq. S6)*
- 1153 *2. replace the earlier shorthand $Q(f_i-f_j)$ with an explicit Normal prior on fitness differences, $\mathcal{N}\left((f_i -$
1154 $f_j) \mid \mu_\delta, \sigma_\delta^2\right)$ (Eq. S6)*

1155 *This Normal prior (formerly Q) encodes the distribution of fitness differences between parent-daughter*
1156 *karyotypes. .*

1157 5.11 Reviewer 1 Minor Point 6

1158 **Are the fitness estimates f used in the binomial logistic regression in figures 4E-F estimated**
1159 **from all longitudinal samples or just the samples before the novel karyotype emerged? If the**
1160 **latter, it would be good to observe a comparison with the former.** *The fitness estimates used for*
1161 *predicting novel karyotype emergence (now Fig 3) are computed using only the data prior to the emergence*
1162 *time point being predicted. This design choice avoids look-ahead bias and ensures that predictions reflect*
1163 *information that would have been available in a real-time forecasting setting. We have clarified this in the*
1164 *figure legend: “Fitness estimates used here are computed using data prior to the point of emergence being*
1165 *predicted to avoid look-ahead bias.” We agree that comparing this to fitness estimates computed from all*
1166 *available timepoints would be of interest. However, including future data in the training set would inflate*
1167 *the predictive power of fitness and confound interpretation—since ALFA-K would essentially be learning*
1168 *from the karyotypes whose emergence it is supposed to predict. For this reason, we restricted our evaluation*
1169 *to the more stringent and unbiased forecasting setup.*

1170 5.12 Reviewer 1 Minor Point 7

1171 **Line 253 – I wouldn’t say that ‘CNAPs across the two untreated lineages exhibited much**
1172 **more evidence of correlation than CNAPS compared between either untreated lineage and**
1173 **the cisplatin treated lineage’.** *In both 5F and 5G, the mean pearson correlation is very close*
1174 *to zero for each Manhattan distribution and the distribution of the individual comparisons*
1175 *is relatively broad.* *We agree and thank the reviewer for pointing this out. The original statement*
1176 *comparing correlation strengths has been removed as it could be misleading given the data distribution.*
1177 *The analysis in this section was substantially revised; the updated approach uses GLMMs to compare*
1178 *fitness effect variance and magnitude across conditions and a linear mixed model to analyze the decay*
1179 *rate of karyotype similarity with distance, providing a more statistically robust assessment of the fitness*
1180 *landscapes. This analysis confirms a high similarity within trajectories from the same cell line/PDX, which*
1181 *falls sharply with karyotype distance. By contrast, karyotypes from unrelated lineages exhibited almost no*
1182 *baseline similarity.*

1183 5.13 Reviewer 1 Minor Point 8

1184 **Why does the maximum number of chromosomes altered vary between cell line and WGD**
1185 **population in figure 6B? Is this due to a limit on how far from the observed population**
1186 **ALPHA-K can estimate karyotype fitness? This should be clarified.** *In ALFA-K we only*

1187 *interpolate fitness for karyotypes within a Manhattan distance of two mis-segregations from the observed*
1188 *frequent clones (Supplementary Section 1.4, also now highlighted in Fig. 1). Since the WGD- subclones*
1189 *in our data exhibit fewer chromosome changes overall compared to the WGD+ subclones, the inferred*
1190 *landscape does not extend as far into regions with many alterations for the WGD- population. Consequently,*
1191 *their plotted maximum number of altered chromosomes naturally appears lower in (what is now) Fig. 5.*
1192 *We have added a brief note to the Results section accompanying Fig. 5 to clarify this.*

1193 **5.14 Reviewer 1 Minor Point 9**

1194 **It is worth mentioning the limitation of only considering total copy number in the discussion.**
1195 **For example, in the context of an inactivating tumor suppressor gene mutation, a $1+1 \rightarrow 1+0$**
1196 **event where the unaffected allele is lost will have a very different fitness effect to a $2+0 \rightarrow 1+0$**
1197 **loss. We agree this is an important limitation. We have added a paragraph to the Discussion acknowledging**
1198 **this and suggesting how ALFA-K could be extended to handle allele-specific data: “A further limitation is**
1199 **that our current implementation treats each chromosome homogeneously, ignoring which allele is lost or**
1200 **gained...”**

1201 **5.15 Reviewer 1 Minor Point 10 (Typos)**

1202 **Line 29 - Should “gene dosage” be “gene doseage”?** *Corrected. Equation 2 – v_{it} should be*
1203 *v_{it} . This equation was part of the old methods section and has been removed/replaced in the revised methods.*
1204 **Line 226 – ‘signficant’.** *Corrected*

1205 **5.16 Reviewer 1 Code Availability**

1206 **I have checked the repository and found that the code for ALPHA-K and the agent-based**
1207 **modelling is present. There is a minimal README but probably not enough to run either**
1208 **of the tools easily. It would be good to document the format for the input files needed to**
1209 **run ALPHA-K and how to interpret the output. We have wrapped ALFA-K into an install-ready**
1210 **R package (<https://github.com/Richard-Beck/alfakR>), added continuous-integration unit tests,**
1211 **and provided a vignette + README that fully document input formats, automatic dependency installation**
1212 **(including the previously missing lhs), step-by-step runs for both ALFA-K and the ABM, and interpretation**
1213 **of all output files.**

1214 **5.17 Reviewer 2: Summary/Flow**

1215 **Regarding flow of the paper, I suggest the authors consider a slightly different ordering. Per-**
1216 **haps the results should come first as many of the methods become clearer through example**
1217 **(at least for this reviewer). Methods could then be presented at the end of the paper with**
1218 **additional details in the SM. If this does not fit the authors’ style, I strongly suggest moving**
1219 **some aspects of the results and diagrams/aspects of diagrams from results into methods -**
1220 **to make methods more understandable and concrete. At a minimum, referring to diagrams**
1221 **within the results would likely aid readers (especially Figure 1). If the authors produced**
1222 **a more detailed methods section (for the SM likely) they also could include an expanded**
1223 **visualization and discussion of the inference and prediction matrices, the definition of nearest**
1224 **neighbors and other important concepts that underlie their interesting methodology. We agree**
1225 **with the benefits of changing the ordering as suggested by the reviewer and have implemented the following**
1226 **structural and content changes: a) The Supplementary Information now contains an expanded and detailed**
1227 **description of the ALFA-K methodology b) To aid readers, the main text now includes Figure 1, which**
1228 **provides an overview of the ALFA-K workflow and conceptualization, directly addressing the reviewer’s**

suggestion regarding the utility of diagrams like Figure 1. Furthermore, the expanded Supplementary Information includes additional figures illustrating the validation procedures and results on synthetic data. c) Concepts such as 'one-MS-step neighbors' are now explicitly defined and detailed within the Supplementary Information Methods, clarifying important concepts as requested. d) The main Methods section has been moved to the end of the manuscript (after the Discussion) and now serves as a high-level overview directing readers to the detailed descriptions in the Supplementary Information. e) Most Figures detailing the validation on synthetic data have been moved to the Supplementary Information and expanded. The main text now focuses on the validation using experimental data, presented earlier in the Results.

5.18 Reviewer 2: More detail

Finally, as noted in more detail below, the paper would benefit from additional detail at focused points throughout. I especially want to highlight adding additional details of simulations run in the methods section, including a description of the biological data used in the results, and more strongly tying the results into the discussion in particular to aid in highlighting how the results from the novel method may also be novel, hypothesis generating results for the biology. We have included details of the simulations in the Supplementary Information (Supplementary Section 2). The description of the biological data is also under Supplementary Information (Supplementary Section 4). Table S1 summarizes the parameter definitions and values used for the ABM. The methods section retains a high level summary of simulation details and biological data used. We have also worked to better connect results to discussion points (e.g., discussion on WGD, CIN/error threshold, limitations).

5.19 Reviewer 2 Line-by-line

11 - "not-yet seen in the data" is confusing to me. This sentence may benefit from reworking. Does it mean "enabling the prediction of emergent karyotypes from input data"? Reworded to: "This method utilizes longitudinal single-cell karyotype data from evolving cell populations to estimate the fitness of thousands of karyotypes near those observed, enabling prediction of emergent karyotypes not yet present in the input data."

29 - Inconsistent use of "mis-segregations" versus "missegregations" (see line 91 for example). Standardized to "missegregation" throughout.

29 - is "gene dosage" a typo? Yes, corrected to "gene dosage".

32 - missing space. Corrected ("In addition to these intracellular effects")

38-40 - Sentence reads in a confusing fashion. Maybe "do not independently..." instead of "are not independently"? Reformulated (to: "Support for the role of genomic context comes from findings that CNAs lacking independent prognostic value can predict survival in combination [15], as well as from the reproducible temporal ordering of CNAs in patient-derived xenograft and organoid models [16, 17].")

41 - Maybe helpful for reader to call out the difference between chromosomal instability and aneuploidy and/or various kinds of chromosomal instability and the process(es) by which it causes aneuploidy. Literature review - Authors maybe interested in this article - which takes a different approach to exploring CIN: <https://elifesciences.org/articles/69799>. We have revised the Introduction (Paragraph 1 and Paragraph 3) to clarify the distinction between aneuploidy and CIN, briefly mention the diverse mechanisms underlying CIN, and explicitly state our focus on whole-chromosome CIN. We have included citations to relevant literature to support these clarifications as well as recent modeling advances at the intersection of CIN and selection.

64 - Does ALFA-K stand for something? If it does, would you consider laying out what it stands for? Thank you for pointing this out. Now defined in abstract.

78 - Typo: LaTeX leading double quote needs used correct quote formatting used throughout

1275 **81 - Likely typo: New paragraph was used mid-sentence.** *This refers to the old methods*
1276 *section, which has been revised. Formatting checked.*

1277 **81- Could the authors define v_0 ? Apologies if I simply missed the definition.** *v_0 (representing*
1278 *initial abundances) was part of the old methods description. The revised methods (Supplementary Section*
1279 *1) define initial frequencies $x_{0,i}$ which are jointly optimized.*

1280 **77-87 - This reviewer is somewhat lost here. Would the authors consider laying out**
1281 **the story of the equations with more detail given how critical this is their method? If**
1282 **appropriate, detail could be added to the supplement, allowing the interested reader to go**
1283 **there for more information/exposition.** *We have significantly expanded the Methods section in*
1284 *the Supplementary Information (Supplementary Section 1) with more detailed explanations, step-by-step*
1285 *descriptions, interpretations, and assumptions for each part of the inference process.*

1286 **Line 79 - The authors define a matrix U but never use it. Is it U hat?** *This refers to the old*
1287 *methods section. The matrix U is no longer defined or used in the revised methods. We now work directly*
1288 *with counts y_{it} and frequencies x_{it} .*

1289 **Line 85 - It might be clearer to say “Defining S to be the set of indices belonging to**
1290 **frequent karyotypes”?** *Agreed. (Supplementary Section 1.2).*

1291 **Line 89 - What does von Neuman (I believe the “v” isn’t capitalized) neighbor mean in**
1292 **the context of this non-spatial model? Could the authors please explicitly call out what they**
1293 **mean? I am guessing von Neuman karyotypes are karyotypes that differ by one count in one**
1294 **dimension in a 22 member vector continuing counts of each chromosome. Is that correct?**
1295 *That is correct. We have replaced the term “Von Neumann neighbors” with one-MS-step neighbors, defined*
1296 *(Supplementary Section 1.3) “karyotypes that differ by a single chromosomal gain or loss”.*

1297 **Line 93 - For clarity, please consider calling out Equation 4 as “Equation 4” or something**
1298 **similar rather than simply “4”.** *Corrected*

1299 **Line 100 - Please capitalize “Gaussian”.** *Ensured correct throughout.*

1300 **Line 101 - As “fields” is a common noun, consider calling out the package name with**
1301 **quotes or in ‘code’ type font.** *`fields` used whenever this package is referred to*

1302 **101 - “von Neuman”.** *See response to “Line 89”*

1303 **111 - “data set” is used rather than “dataset”. The paper typically uses “dataset”. Please**
1304 **consider updating line 111. Dataset used consistently throughout.** *“Data set” ensured absent*
1305 *in revision.*

1306 **113 - Would the authors define R^2 either in line or referring the reader to where its defined**
1307 **in the text?** *This refers to the old methods section. R^2 (now R_X^2 for cross-validation) is now defined*
1308 *explicitly with its formula in the Supplementary Information Methods Eq. S11) and its interpretation is*
1309 *discussed in the context of performance evaluation (Supplementary Section 3.1).*

1310 **116 - In this section, could the authors lay out explicitly how the ABM is used in the**
1311 **inference process? It appears to be used in the inference process later in the paper, not**
1312 **only for generating synthetic data for testing.** *The ABM is not used in the ALFA-K inference*
1313 *process itself. We have clarified its distinct roles in the revised manuscript. It is used for: 1) Gener-*
1314 *ating synthetic data with known ground-truth fitness landscapes to validate ALFA-K’s inference accuracy*
1315 *(detailed in Supplementary Information Section 3). 2) Simulating future evolution after ALFA-K has in-*
1316 *ferred a landscape from empirical or synthetic data, which is used for forecasting and testing hypotheses*
1317 *(e.g., predicting emergence, predicting future passages, exploring missegregation rate effects). The forward*
1318 *simulation process is detailed in Supplementary Information Section 1.6 (“Predicting Future Evolution:*
1319 *Forecasting Simulation”) and Section 1.6 (“Calculating the Steady-State Distribution”).*

1320 **120 - To aid readers, I suggest the authors add a brief overview of the methods and**
1321 **corresponding per-chromosome missegregation rates in their supplementary material rather**
1322 **than referring out for these critical methods and parameters?** *We agree this aids the reader. In*
1323 *the revised manuscript, Table S1 (Supplementary Information) now summarizes the parameter definitions*
1324 *and values used for both synthetic and biological data.*

1325 **126 - Could the authors please cite literature or otherwise justify the parameters used in**
1326 **their simulations? As appropriate, readers could be referred to SM. Table S1 in the Supple-**
1327 **mentary Information, which summarizes the parameter definitions and values, now also includes a column**
1328 **with references and justification for our respective parameter range choices.**

1329 **Line 136 - Equation 6 - would the authors consider increasing the size of the parentheses**
1330 **for better presentation? Also, please use Roman type font for “sin”. This refers to the old**
1331 **GRF equation. The revised equation defining the GRF landscape (now Eq. S12 in Supplementary Section**
1332 **2) uses sin and appropriate parenthesis sizing for clarity.**

1333 **Line 139 - Could the author explain/expand on the α_{ik} and α_{ij} as vectors given that in**
1334 **index k seems to represent only a single entry in the karyotype space K? To the reviewer,**
1335 **at least it seems like a_{ik} might represent a single dimension in karyotype i rather than the**
1336 **whole 22 dimensional karyotype vector. This refers to the old transition matrix description. We**
1337 **have revised the notation for clarity in the Supplementary Information (1.3, 1.6). We now use α to**
1338 **represent the full karyotype vectors. The probability of transition between these states $P(\alpha_i|\alpha_j)$ and the**
1339 **rate matrix Q used in forecasting (Supplementary Section 1.6) implicitly consider the full vectors and the**
1340 **specific chromosome change involved, rather than relying on the potentially ambiguous α_{ik} indexing in the**
1341 **equations.**

1342 **Line 144 - Inline equation is missing additional index i for variable n. The description of**
1343 **transitions due to missegregation has been revised and clarified (Supplementary Sections 1.3, 1.6), using**
1344 **notation ($P(\alpha_i | \alpha_j)$, $Q_{i \leftarrow j}$) that incorporates the necessary information about the parent karyotype and the**
1345 **specific chromosome change without the ambiguity of the previous inline formula.**

1346 **Line 146 - Would the authors consider specifying N_i in words? I believe its the total**
1347 **number of chromosomes in karyotype i? Yes, N_i in the old manuscript referred to the total number**
1348 **of chromosomes (ploidy) of karyotype i. In the revised manuscript, while the concept of ploidy is discussed,**
1349 **particularly in the Results section (Fig. 6) and Discussion, the symbol N_i is no longer used within the**
1350 **core method equations in the Supplementary Information (Supplementary Section 1). We specify ploidy in**
1351 **words where relevant to the interpretation.**

1352 **Line 150 - This appears to be a great summary of the methods. As noted in my summary**
1353 **comments, bringing this summary up earlier in the manuscript would benefit readers as well**
1354 **as including the paragraphs below. In some sense, the initial “results” included may be**
1355 **considered ”methods” in the sense that these first results illustrate and validate the method.**
1356 **Figure 3 could be reconceived as the first result - when the method meets the non-synthetic**
1357 **data. We agree and have restructured our manuscript as described in detail in response to Reviewer #2**
1358 **SummaryFlow:**

1359 **Figure 1: B - Could the authors include the value of lambda for this example? D: If each**
1360 **karyotype can only change by one missegregation, how do agents arise that differ by more**
1361 **than one chromosome? Are there transient, low frequency populations not included? This**
1362 **panel no longer exists following significant manuscript restructuring in response to other comments. That**
1363 **is a simplifying assumption made during the landscape reconstruction - ABM agents can undergo multiple**
1364 **simultaneous missegregations as clarified in the updated description (“At division, each chromosome copy in**
1365 **a dividing cell may independently mis-segregate with probability p, relocating both copies of that chromosome**
1366 **to a single randomly chosen daughter cell”).**

1367 **172 - Would the authors please explaindescribe which simulationscomputational exper-**
1368 **iments they performed? Like - the values of lambda used, the number of stochastic replicates**
1369 **per experimental condition, etc. Table S1 now summarizes the parameters used for synthetic data**
1370 **generation.**

1371 **176 - Could the authors please explain what specifically they mean by number of longitu-**
1372 **dinal samples? I assume its the number of stochastic replicates? The ”number of longitudinal**
1373 **samples” refers to the number of distinct time points at which the evolving population was sampled within a**
1374 **single simulation, not the number of replicate simulations. To clarify this, we have revised the text to now**

1375 read: "...The validation showed that inference accuracy, primarily measured by Spearman's correlation,
1376 improved with smoother landscapes and an increased number of sampled time points..."

1377 **183 - Would be helpful to explain angle metric inline.** *This is a helpful suggestion. To improve*
1378 *readability, we have added a brief, intuitive explanation of the angle metric directly in the text where it*
1379 *is first mentioned. The revised sentence now reads: "We used the angle metric (a measure of similarity*
1380 *between two evolutionary trajectories, where an angle of 0 degrees indicates identical paths of change) to*
1381 *evaluate predictive performance..."*

1382 **185 - Could the authors provide an interpretation of a score and/or a landscape that**
1383 **produces a R^2 score that is positive versus 0 versus negative? I assume it is simply that**
1384 **the algorithm fails to characterize these landscapes (at least with the given input stochastic**
1385 **replicates)?** *We have a more in-depth characterization of different landscape metrics in the restructured*
1386 *supplement, in particular Figs. S2F and S3B, showing how landscape complexity, number of sampled*
1387 *timepoints and other features of the training data influence the R^2 score.*

1388 **Figure 2: E: I suggest the authors consider increasing the alpha value or adding a darker**
1389 **border to the individual sample points. The points in yellow end up presented as rather**
1390 **fuzzy, making it difficult to see the boundaries of the distribution. Also, what is meant by**
1391 **the rows frequent, distance 1 and distance 2? I do not see those designations called out.**

1392 *In the revision we have opted to represent data quantiles rather than attempt to show all data points, in*
1393 *order to mitigate this issue. These represent the karyotypes with fitness estimates from different steps of*
1394 *the pipeline (frequent \rightarrow neighbours \rightarrow kriging). We have been more consistent with this terminology in*
1395 *the revision to aid clarity.*

1396 **194 - Could the authors please provide more background for this data? What is SA532?**
1397 **What are the relevant experimental conditions that generated these data? How many lineages**
1398 **are in the data set and what are they from?** *We have included a supplementary section 4 to convey*
1399 *various aspects about the data such as whether they are from PDX or cell line, treated vs untreated, etc.*

1400 **197-198 - Something is off with the sentence beginning "Thus e. g. ...". Consider starting**
1401 **with "For example, in SA532 ... "?** *In our revised manuscript we hope the text corresponding to*
1402 *this sentence reads more clearly ("Here, a trajectory is the ordered sequence of passages obtained from a*
1403 *single lineage (e.g. $A \rightarrow B \rightarrow C$)...")*

1404 **200-201 - For clarity, please consider adding commas after "found" and "results".** *Reworded*
1405 *"Examining R^2 metric scores for all lineages obtained from the data, we found in agreement with our ABM*
1406 *results that longer lineages result in better fits" to: "In line with in silico tests, CV scores generally increased*
1407 *with the number of training passages used".*

1408 **206 - Missing space in text.** *Corrected.*

1409 **Figure 3: C and D captions: Could the authors call out ECDF in the caption so that the**
1410 **caption matches the diagram?** *In the corresponding figure in the revision (Fig. 2C), we now use*
1411 *"cumulative frequency" distribution in both figure panel and caption.*

1412 **216 - 217 - This appears to be a run-on sentence.** *Reworded for clarity.*

1413 **Figure 4: Caption: A) what is clonal interference? Could the authors explain what that**
1414 **means in this context and how its related to the experiment? Overall, using the caption and/or**
1415 **text could the authors relate connect 4A back to the text? There is a typo in the second**
1416 **line.** *We thank the reviewer for this helpful suggestion. In the revised Fig. 3 caption, we have defined*
1417 *clonal interference more clearly and explained its role in the context of our experiment. Specifically, we*
1418 *now clarify that clonal interference refers to the competition between distinct karyotype lineages, each with*
1419 *potential to expand, but which may hinder each other's fixation over time. This phenomenon is visualized*
1420 *in panel A, where multiple candidate karyotypes expand concurrently but do not all dominate. The "}" noted*
1421 *in the second line is not a typo but refers to the set of karyotypes that are absent from ζ (i.e. ' ζ '). To*
1422 *reinforce clarity, we added a brief statement in the main text connecting clonal interference in panel A to*
1423 *the prediction of emerging karyotypes. These changes should better communicate the relevance of Fig. 3*
1424 *within the framework of forecasting karyotype evolution.*

1425 **231 - PDX datasets have not yet been discussed at this point. Could the authors discuss**
1426 **what data sets they are working as well as define the abbreviation?** *We thank the reviewer*
1427 *for pointing this out. In the revised manuscript, we now define the PDX abbreviation at first mention and*
1428 *briefly describe the data source. Specifically, we clarify that the analysis includes both breast cancer cell*
1429 *lines and patient-derived xenograft (PDX) models subjected to serial transplantation, as originally reported*
1430 *by Salehi et al. We also added a pointer to the detailed description in the Data Source section.*

1431 **234 - Which conditions?** *The sentence has been revised: "...We set out to quantify how experimental*
1432 *context (in vitro vs. PDX) and cisplatin exposure reshape the fitness landscapes inferred with ALFA-K."*

1433 **243 - There is an exclamation point rather than an expected symbol.** *Thank you for catching*
1434 *this typo. The exclamation point should be a "less than" symbol. This has been corrected.*

1435 **254 - What quality control? Is this described in the manuscript?** *The quality control check*
1436 *is the cross-validation procedure described in the Methods section. We have amended the text to make this*
1437 *explicit.*

1438 **Figure 5: Pearson needs capitalized. Manhattan needs capitalized.** *We have corrected the*
1439 *capitalization of "Pearson" and "Manhattan" in the Figure legends and throughout the manuscript.*

1440 **290 - "dependent" is misspelled.** *"dependend" has been corrected to "dependent."*

1441 **340 - 346 - Could the authors tie this narrative back to a specific set of their results?** *We*
1442 *substantially reworded the results section accompanying 6 to more closely tie our results to this discussion*
1443 *point regarding "quasispecies".*

1444 **341 - Typo: " needed.** *Thank you for catching this formatting error. The double backticks have*
1445 *been corrected to standard quotation marks.*

1446 **359 - Could the authors expand on what they mean by interferon gamma as a metric of**
1447 **missegregation? In what context is interferon gamma a metric of this?** *We previously used In-*
1448 *terferon Gamma Signaling as a surrogate measure of chromosome missegregations (Kimmel & Beck et al.,*
1449 *2023). Chromosome missegregations can trigger the formation of micronuclei, which, upon rupture, release*
1450 *genomic DNA into the cytosol (Bakhoun et al., 2018). Cytosolic dsDNA is detected by the cGAS-STING*
1451 *pathway (Sun, Lijun and Wu et al., 2013), resulting in the induction of type I interferon-stimulated genes*
1452 *(Lan, Yuk Yuen and Londoño et al., 2014; Mackenzie et al., 2017). This process leads to the upregulation*
1453 *of interferon production, which subverts otherwise lethal epithelial responses to cytosolic DNA. We have*
1454 *added a brief explanation of this to the discussion.*

1455 **368 - Typo: estimated** *Thank you for this correction. We have added the missing "d".*

1456 5.20 Reviewer 2 Code Availability

1457 **I have checked the repository and found that the code for ALPHA-K and the agent-based**
1458 **modelling is present. There is a minimal README but probably not enough to run either**
1459 **of the tools easily. It would be good to document the format for the input files needed to run**
1460 **ALPHA-K and how to interpret the output. Code: Well organized repository with extensive**
1461 **READMEs and documentation of figures in paper Code downloads and compiles. Generates**
1462 **warning: In file included from .ABMmain.cpp:17: .ABMsetup.h:6:26: warning: implicit**
1463 **conversion from 'long' to 'int' changes value from 1000000000 to 1410065408 [-Wconstant-**
1464 **conversion] int $pop_{write_freq} = 1000000000$; Runs until Process output from ABM simulation**
1465 **Get error that $proc_{sim}$ isn't available. Similar error for $alfak$ function. Demonstrated code**
1466 **compiles and at least a good portion runs. Note I didn't try to troubleshoot - just running as**
1467 **is. Additional investigation showed that as expected, the functions $proc_{sim}$ and $alfak$ are in**
1468 **the R - just for some reason not pulling up on my machine. Perhaps something is missing in**
1469 **README in this section?** *We thank the reviewer for their positive feedback on the code's availability*
1470 *and organization. We have wrapped ALFA-K into an install-ready R package ([https://github.com/Richard-](https://github.com/Richard-Beck/alfakR)*
1471 *Beck/alfakR), added continuous-integration unit tests, and provided a vignette + README that fully*
1472 *document input formats, automatic dependency installation (including the previously missing lhs). See*

1473 *also <https://github.com/Richard-Beck/ALFA-K> for step-by-step runs for both ALFA-K and the ABM used*
1474 *in this work.*

REVIEWERS' COMMENTS (point by point response)

Reviewer #2 (Remarks to the Author):

Revision - summary:

The authors substantially and thoroughly updated the manuscript, effectively touching all parts of the work. It is much clearer now - with the substantial methods plainly laid out and easily followed. The overall flow is greatly improved and the work highlights the application of its methods for hypothesis generation. It is clear which biological problem is addressed, why it is important to understand, how this method addresses it, and what it may provide to the community. The authors make methods reproducible through their detailed descriptions in the paper and thorough R package, READMEs, and a comprehensive worked example.

Small comments:

Title: What is adaptive about this? Its late - but consider if that's the best description. Is it adaptive mapping or mapping of the adaptive landscape? At a glance, it doesn't feel like it's the mapping that is adaptive, but rather the method maps the adaptive fitness landscape.

We appreciate this feedback regarding clarity. We have updated the title to 'ALFA-K: Local Adaptive Mapping of Karyotype Fitness Landscapes' at the specific request of the Editor. Scientifically, we believe 'Adaptive Mapping' is appropriate because the algorithm is not static; it adaptively defines the local mapping domain around the specific clones observed in a sample (rather than fitting a fixed global function) and adapts its fitness estimates to the specific competitive dynamics of each lineage.

Results

Figure 1 (and throughout): Caption - F: The caption refers to colored tick marks as marking the time windows used for training and grey dots indicating the final training time point. This reviewer sees grey ticks rather than grey dots. Also, given that there are multiple colored ticks showing the time windows, it is unclear which segments are sampled and which are not. Is the sample between yellow and green, then sampled again between purple and gray? I suggest clarifying.

We thank the reviewer for identifying these inconsistencies. We have corrected the captions in Figure 1F and Supplementary Figure S1E to accurately describe the symbols as 'ticks' rather than 'dots' (including the red ticks marking the prediction horizon). We also clarified the definition of the time windows to resolve ambiguity: colored ticks indicate the initial timepoint of a training window, while grey ticks indicate the final timepoint.

First results section: Suggest brief explanation of the metric produced by cross-validation and procedure. Since many metrics could be produced by CV (such as a mean-squared error), at least something like "We use the XXX metric to assess model fit in CV" and would consider making sure the reader knows "higher is better" (as opposed to lower is better in the MSE) and

the metrics bounds. I also suggest noting that you use leave 1-out CV, predicting the fitness of the left out samples. Finally, I am not sure that the abbreviation CV is ever defined - but I easily may have missed it.

We have updated the Results section to explicitly define 'CV' upon first use as **leave-one-out cross-validation**, noting that it yields a score (max 1.0) where higher values indicate better fit. To improve clarity, we also removed the undefined mention of 'CV scores' from the Figure 1 legend, referring instead to 'internal consistency' until the metric is formally introduced.

First results section - minor comment: In the final paragraphs, the narrative becomes more challenging to follow. It is understood that the authors are covering a conceptually challenging section of their results (at least to this reviewer) and I can see the content is all there. Sometimes things are just dense due to concept and need for brevity. That said, it was challenging for this reviewer to parse through it. There may not be room for change here and I do not request any, but pass it along if it may help the authors.

To improve readability, we have simplified the sentence structure in the final paragraphs of this section, specifically separating the methodological definitions from the results.

Methods

Equation 2: Something seems off. Spacing between terms for sure and there may be an operator or term missing between $x_j(t)$ and f_i . Also, the authors may not intend to have an indent (new paragraph) in the sentence that follows. I noticed this same thing in the SM as well (will comment directly there as well). Finally, I suggest using a period at the end of Equation 2, assuming the sentence that follows is a new sentence.

Thank you for pointing these errors out. We have added the missing '+' operator and addressed the formatting issues.

Supplement

Equation S1: I am guessing that a new paragraph (indent) is not intended after S1. I am guessing there is an extra return in the LaTeX source after equation S1. This occurs throughout.

Thank you for pointing this out. We have corrected this indentation issue throughout.

End of SM S1: Many equal signs separating sections. **Removed**

Typos/inline comments:

Line 290 - consider expanding out cGAS-STING - unless that is truly the term of the art.

We thank the reviewer for this suggestion. To ensure accessibility for a broad readership, we have expanded the acronym at its first mention (cyclic GMP-AMP synthase–stimulator of interferon genes).

Line 79 - extra space after ALFA; Line 357 - normal; Line 874 - Euclidean. All addressed.

Reviewer #2 (Remarks on code availability):

Reviewed two repositories. Spot checked that it contains the expected languages (C++ and R) and sensible syntax. Appears to be complete. Repository appears to be thoroughly annotated with READMEs as well as scripts to run both the complete results of the paper as well as a separate example, positioned as a locally installed R package.

Reviewer #3 (Remarks to the Author):

This study demonstrates a computational framework (ALFA-K) to determine the fitness landscape for varying karyotypes based on time-series scWGS data. This framework is thoughtfully designed, and its application to data from the Shah lab yields valuable insights into the context-specific effects of aneuploidy in cancer. Nevertheless, I suggest several points for improvement:

Assumption on pre-existing karyotypes:

The method assumes that all frequent karyotypes are pre-existing. This raises questions about the simulation setup. Specifically, how the initial fractions of these karyotypes are determined and whether they can be accurately estimated from synthetic data. It would be helpful to report the accuracy of initial fraction estimation derived from joint likelihood optimization.

We thank the reviewer for raising this important point regarding the model's assumptions. While ALFA-K models frequent karyotypes as growing exponentially from $t=0$, this is a mathematical framework designed to accommodate late-emerging clones by fitting them with a negligible initial frequency and an appropriate growth rate.

To demonstrate that this assumption does not bias the results, we validated the initial fraction estimation using our synthetic dataset (see: "Evaluating Fitness Landscape Inference Accuracy"). We directly compared the inferred relative frequencies at the start of the sampling window against the observed frequencies in the synthetic samples. We found:

1. High Accuracy: The inferred initial fractions matched the ground truth with high precision (Pearson correlation > 0.99).
2. Robustness to Late Arisers: For karyotypes that had not yet emerged in the sample at the start of the window (True Frequency = 0), ALFA-K correctly estimated negligible abundances (Median $< 0.02\%$).

This confirms that the joint likelihood optimization accurately recovers the initial population structure and is not confounded by late-emerging clones.

Effect of missegregation rate on fitness inference:

It would be useful to assess whether increasing the missegregation probability in the agent-based model affects the accuracy of fitness inference. In particular, transitions between frequent karyotypes could potentially obscure or confound the inferred fitness landscape.

We agree that transitions between karyotypes are dynamically important, particularly during the initial emergence of a clone. However, ALFA-K is designed to infer fitness from the growth phase of **frequent** karyotypes, a regime where intrinsic growth mathematically dominates mutational influx.

To demonstrate this, consider the rate of change for a clone i receiving influx from a parent j :

$$\frac{dx_i}{dt} \approx f_i x_i + \mu_{j \rightarrow i} f_j x_j$$

The relative contribution of influx, $R(t)$, evolves over time as the clone expands:

$$R(t) = \frac{\text{influx}}{\text{growth}} = \mu_{j \rightarrow i} \left(\frac{f_j}{f_i} \right) \left(\frac{x_j(t)}{x_i(t)} \right)$$

Assuming competing frequent clones have comparable fitness ($f_i \approx f_j$), this simplifies to:

$$R(t) \approx \mu_{j \rightarrow i} \left(\frac{x_j(t)}{x_i(t)} \right)$$

This ratio reveals two distinct dynamical regimes:

1. **Emergence Phase** ($x_i \ll x_j$): Initially, when a clone is rare, $R(t)$ is large, and dynamics are driven by influx.
2. **Establishment Phase** (x_i becomes frequent): As the clone expands, the ratio x_j/x_i drops rapidly. Once clone i reaches the "frequent" threshold used by ALFA-K (e.g., comparable abundance to j), the ratio $R(t)$ asymptotically approaches $\mu_{j \rightarrow i}$.

For a biologically high missegregation rate $p \approx 10^{-3}$, the specific transition rate $\mu_{j \rightarrow i}$ is on the order of 10^{-4} - 10^{-5} (since missegregations are distributed among chromosomes: only a fraction can result in a specific transition $i \rightarrow j$). Thus, as soon as clone i becomes established $x_i \approx x_j$, the contribution of influx drops to $\approx 0.001\%$. Even under pessimistic scenarios, e.g. if the parent j remains 100 times more abundant, influx contributes only $\approx 0.1\%$ to the growth rate.

Crucially, our experimental data imposes a detectability threshold that effectively confines our analysis to this "Establishment Phase." With typical single-cell sample sizes of hundreds of cells per timepoint, a clone must reach a frequency of $\approx 0.1\%$ to be reliably detected and classified as "frequent." By the time a clone crosses this visibility threshold, the abundance ratio x_j/x_i is already sufficiently low that mutational influx is negligible. Furthermore, because ALFA-K's inference methods (both the initial least-squares estimation and subsequent likelihood maximization) are inherently weighted by clone abundance, the fit is overwhelmingly driven by the high-frequency data points where the signal-to-noise ratio is highest and intrinsic growth dominates.

Consequently, while we utilized a conservative baseline of $p = 10^{-5}$ for our primary synthetic benchmarks to ensure computational tractability at the scale of our simulations, this analysis confirms that the reported accuracy holds even for biologically high-CIN scenarios. This robustness is empirically supported by our sensitivity analysis (see **Supplementary Figure 3**), where inference accuracy remained stable even at elevated error rates.

Details on growth offset correction:

ALFA-K models the frequency of karyotypes by an ODE, assuming the population follows the exponential growth model. To infer the fitness of frequent karyotypes, the author conducts a two-step optimization process: first solving a QP, followed by maximum likelihood estimation. This approach is appropriate for fraction data. However, in the subsection "Growth Offset Correction for Passaging Experiments", the manuscript states that predicted fitness values can be shifted to obtain an exponential growth rates. However, this shift is not obvious, and it might be beneficial to provide more details here to clarify.

We thank the reviewer for this suggestion. We have clarified in the Supplementary Methods ("Growth Offset Correction for Passaging Experiments") that the replicator equation depends strictly on fitness *differences* relative to the mean ($f_i - \bar{f}$). Because the equation is invariant to additive shifts, the optimization in Step 1 yields a **relative** fitness vector. The 'growth offset correction' is simply the process of anchoring this vector by adding a global constant ensuring that the mean fitness matches the experimentally observed population growth rate without altering the fitted frequency dynamics.

Minor comments:

The equation S3 has a typo in objective function, where r seems undefined. Corrected $r \rightarrow c$

For notation in model definition, it might be helpful to define the initial frequencies as $x_{i,0}$ to make it consistent with definition of other $x_{i,t}$. In the section "Refinement via Joint Likelihood Optimization" we have corrected several instances where the indices were in the wrong order. We now consistently use

$x_{\text{species,time}}$

In page 8 line 190, a typo (Fig 4E -> Fig 4F) corrected

Reviewer #4 (Remarks to the Author):

Reviewer #4 (Remarks on code availability):

The code is executable.

Revision - summary:

The authors substantially and thoroughly updated the manuscript, effectively touching all parts of the work. It is much clearer now - with the substantial methods plainly laid out and easily followed. The overall flow is greatly improved and the work highlights the application of its methods for hypothesis generation. It is clear which biological problem is addressed, why it is important to understand, how this method addresses it, and what it may provide to the community. The authors make methods reproducible through their detailed descriptions in the paper and thorough R package, READMEs, and a comprehensive worked example.

Small comments:

Title: What is adaptive about this? Its late - but consider if that's the best description. Is it adaptive mapping or mapping of the adaptive landscape? At a glance, it doesn't feel like it's the *mapping* that is adaptive, but rather the method maps the adaptive fitness landscape.

Results

Figure 1 (and throughout): Caption - F: The caption refers to colored tick marks as marking the time windows used for training and grey dots indicating the final training time point. This reviewer sees grey ticks rather than grey dots. Also, given that there are multiple colored ticks showing the time windows, it is unclear which segments are sampled and which are not. Is the sample between yellow and green, then sampled again between purple and gray? I suggest clarifying.

First results section: Suggest brief explanation of the metric produced by cross-validation and procedure. Since many metrics could be produced by CV (such as a mean-squared error), at least something like "We use the XXX metric to assess model fit in CV" and would consider making sure the reader knows "higher is better" (as opposed to lower is better in the MSE) and the metrics bounds. I also suggest noting that you use leave 1-out CV, predicting the fitness of the left out samples. Finally, I am not sure that the abbreviation CV is ever defined - but I easily may have missed it.

First results section - minor comment: In the final paragraphs, the narrative becomes more challenging to follow. It is understood that the authors are covering a conceptually challenging section of their results (at least to this reviewer) and I can see the content is all there. Sometimes things are just dense due to concept and need for brevity. That said, it was challenging for this reviewer to parse through it. There may not be room for change here and I do not request any, but pass it along if it may help the authors.

Methods

Equation 2: Something seems off. Spacing between terms for sure and there may be an operator or term missing between $x_j(t)$ and f_i . Also, the authors may not intend to have an indent (new paragraph) in the sentence that follows. I noticed this same thing in the SM as well

(will comment directly there as well). Finally, I suggest using a period at the end of Equation 2, assuming the sentence that follows is a new sentence.

Supplement

Equation S1: I am guessing that a new paragraph (indent) is not intended after S1. I am guessing there is an extra return in the LaTeX source after equation S1. This occurs throughout.

End of SM S1: Many equal signs separating sections.

Typos/inline comments

Line 79 - extra space after ALFA

Line 290 - consider expanding out cGAS-STING - unless that is truly the term of the art.

Line 357 - normal

Line 874 - Euclidean